# Global, cell non-autonomous gene regulation drives individual lifespan among isogenic *C. elegans*

Holly E Kinser[1,2], Matthew C Mosley[2,3], Isaac B Plutzer[2], Zachary Pincus[2]*

[1]Department of Biomedical Engineering, Washington University in St. Louis, St. Louis, United States; [2]Department of Developmental Biology and Department of Genetics, Washington University in St. Louis, St. Louis, United States; [3]Program in Developmental, Regenerative, and Stem Cell Biology, Washington University in St. Louis, St. Louis, United States

**Abstract** Across species, lifespan is highly variable among individuals within a population. Even genetically identical *Caenorhabditis elegans* reared in homogeneous environments are as variable in lifespan as outbred human populations. We hypothesized that persistent inter-individual differences in expression of key regulatory genes drives this lifespan variability. As a test, we examined the relationship between future lifespan and the expression of 22 microRNA promoter:: GFP constructs. Surprisingly, expression of nearly half of these reporters, well before death, could effectively predict lifespan. This indicates that prospectively long- vs. short-lived individuals have highly divergent patterns of transgene expression and transcriptional regulation. The gene-regulatory processes reported on by two of the most lifespan-predictive transgenes do not require DAF-16, the FOXO transcription factor that is a principal effector of insulin/insulin-like growth factor (IGF-1) signaling. Last, we demonstrate a hierarchy of redundancy in lifespan-predictive ability among three transgenes expressed in distinct tissues, suggesting that they collectively report on an organism-wide, cell non-autonomous process that acts to set each individual's lifespan.

*For correspondence:
zpincus@wustl.edu

**Competing interests:** The authors declare that no competing interests exist.

## Introduction

Across the tree of life, lifespan varies greatly, not only among species but also among individuals of the same species (*Jones et al., 2014*). Much effort has been devoted to discovering genetic and environmental factors associated with longevity in humans (*Milman and Barzilai, 2016*; *Moskalev et al., 2014*; *Dato et al., 2017*), and many lifespan-extending perturbations in model organisms like *C. elegans* have been discovered (*Kenyon, 2010*; *Kenyon et al., 1993*; *Lucanic et al., 2013*). However, little is known about how differences in lifespan arise among individuals within a population. Studies of identical twins demonstrate that lifespan is not particularly heritable, estimating that at most 30% of the variation in human lifespan can be attributed to genetics alone (*Herskind et al., 1996*; *Ljungquist et al., 1998*; *Skytthe et al., 2003*; *McGue et al., 1993*). Moreover, accounting for assortative mating reduces that figure to less than 10% (*Ruby et al., 2018*). Other factors like shared environment explain very little of the remaining variation between individuals (*Herskind et al., 1996*; *Ljungquist et al., 1998*; *McGue et al., 1993*). A reasonable fraction of the variance in human longevity is thus determined by non-genetic, non-environmental factors, likely of stochastic origin. Understanding what these processes are and how they arise provides insight into why some individuals live longer than others.

While these questions are difficult to study in humans, the self-fertile hermaphrodite *Caenorhabditis elegans* is an ideal model for investigating the role of stochastic events in the aging process.

Even genetically identical *C. elegans* raised in standardized environments have widely different lifespans, displaying a degree of variability similar to that of genetically diverse human populations (*Kirkwood et al., 2005*; *Vaupel et al., 1998*; *Zhang et al., 2016*; *Pincus et al., 2011*). Specifically, the coefficient of variation (CV) of lifespan in inbred *C. elegans* is comparable to that of outbred human populations (CV estimates for *C. elegans* range from 0.19 to 0.23 vs. 0.17–0.21 in humans) (*Zhang et al., 2016*; *Stroustrup et al., 2013*; *Gavrilova et al., 2012*; *Arias and Xu, 2017*). The genetics of *C. elegans* aging are well understood (*Kenyon, 2010*) and its external environment is easily controlled. Furthermore, the short lifespan ($\approx$ 2 weeks at 20°C), the ability to generate clonal, genetically identical populations, and the optically transparent body of *C. elegans* make the organism a tractable model to study how and why differences in lifespan arise.

It is often assumed that differences in lifespan among identical individuals are the result of differences in the random accumulation of damage over time. That is, by chance, some individuals encounter more frequent or injurious assaults from their external and/or internal environment than others, resulting in paths that become increasingly divergent with age. In this model, differences in exogenous or endogenous damage precede late-life biological differences between individuals (e.g. differences in gene expression), and ultimately differences in lifespan.

An alternate hypothesis, however, is that pre-existing biological differences lead some individuals to be more vs. less tolerant to such damage in the first place – which would then produce differences in ultimate lifespan. This might come about by a mechanism such as hormesis, in which individuals exposed to a biological stressor upregulate stress-response programs that persist past the original exposure and confer resistance to future stresses (*Kumsta et al., 2017*; *Lithgow et al., 1995*; *Cypser and Johnson, 2002*). Chance events early in life might cause some individuals but not others to stochastically enter into organismal states of heightened stress-vigilance, effectively committing them to longer future lifespans. Even small, stochastic fluctuations in the activity level of key regulatory genes could become stabilized and amplified by positive feedback loops, locking different individuals into distinct biological states. Indeed, feedback-stabilization of mutually exclusive gene-expression states is a hallmark of fate commitment decisions, from the lysis vs. lysogeny switch in bacteriophage lambda (*Ptashne, 2011*) to lineage commitment in hematopoiesis (*Kato and Igarashi, 2019*) and other cell types (*Wang et al., 2009*; *Ferrell, 2012*).

If long vs. short life is the result of stable differences in the gene-regulatory programs executed by different individuals, then long- and short-lived individuals should be distinguishable early in life by the expression of genes that are regulated by those programs. In other words, we propose that identifying genes whose expression early in life is predictive of future lifespan would provide strong evidence that differences in gene regulation and expression can lead previously identical individuals toward different lifespans.

Indeed, several genes and regulatory processes have been identified whose expression, long before death, predicts future lifespan in isogenic *C. elegans* (*Pincus et al., 2011*; *Sánchez-Blanco and Kim, 2011*; *Rea et al., 2005*; *Bazopoulou et al., 2019*). Expression of *hsp-16.2* after exposure to heat shock correlates with the degree of hormetic lifespan extension afforded by that heat shock (*Rea et al., 2005*). This appears to be due to variable activation of thermosensory neurons after heat shock, leading (via variable IIS activity in intestinal cells) to heat-shock responses of different strengths (*Burnaevskiy et al., 2019*; *Mendenhall et al., 2017*). Likewise, expression of several genes associated with aging and/or IIS, including *sod-3*, a stress response gene often used as a reporter for the activity of the IIS-responsive transcription factor DAF-16, correlate with future lifespan when measured at middle age in unperturbed individuals (*Sánchez-Blanco and Kim, 2011*). Other more phenomenological predictors of lifespan, such as movement (*Zhang et al., 2016*; *Hsu et al., 2009*), size (*Zhang et al., 2016*; *Pincus et al., 2011*), redox state (*Bazopoulou et al., 2019*), and accumulation of autofluorescent material (*Zhang et al., 2016*; *Pincus et al., 2011*; *Pincus et al., 2016*) have also been described.

MicroRNAs (miRNAs), short non-coding RNAs that repress translation of many target transcripts, have been identified as both positive and negative markers of future longevity in *C. elegans*. Expression of *mir-71* and *mir-246*, measured via fluorescence of *promoter*::GFP reporters, positively correlates with future lifespan in isogenic individuals, while expression of *mir-239* negatively correlates (*Pincus et al., 2011*). Genetic manipulation of these miRNAs directly extends (*mir-71*, *mir-246*) or shortens (*mir-239*) lifespan through the insulin/insulin-like growth factor (IGF-1) signaling (IIS) and DNA damage response pathways, demonstrating that genetic predictors of lifespan may also act as

functional determinants of longevity (*de Lencastre et al., 2010*). Other miRNAs have also been shown to both promote or antagonize longevity in *C. elegans* through canonical aging pathways (*Boehm, 2005*; *Smith-Vikos et al., 2014*; *Yang et al., 2013*).

While multiple predictive biomarkers of lifespan have been previously reported, these studies have been limited to small sets of candidate genes, generally hand-picked as reporters of either chronological age or IIS activity. Broadly, much of the work in this area, including our own, can be read to suggest that variability in IIS, which can be read out via (presumably rare) 'biomarkers of longevity', accounts for the bulk of variability in individual lifespan. In this work, we systematically revisit those conclusions by examining a large collection of *promoter*::GFP transgenes for their relationship with lifespan.

To this end, we employed a high-density individual culturing device developed by our lab (*Zhang et al., 2016*; *Pittman et al., 2017*) and automated microscopy and image processing to identify additional miRNA biomarkers of longevity. We chose to examine miRNAs because they regulate gene expression of many targets, they have been previously implicated in aging and lifespan, and, unlike transcription factors, their activity is well-represented by fluorescent *promoter*::GFP reporters. We screened 22 *PmiRNA*::GFP reporter strains and found 10 in which GFP levels robustly predict future lifespan among isogenic *C. elegans*. We used two of the most lifespan-predictive miRNA reporters, *Pmir-47*::GFP and *Pmir-243*::GFP, to investigate the specific pathways underlying stochastic variation in longevity. Our findings demonstrate that unlike most known biomarkers of longevity, these reporters are independent of the *daf-16* branch of the IIS pathway in their ability to predict lifespan; however, the microRNAs *mir-47* and *mir-243* themselves are not functional determinants of lifespan. Analysis of dual reporter strains indicates that there is redundancy between predictive *PmiRNA*::GFP reporters, indicating that they act in a hierarchical manner to report on shared, cell non-autonomous lifespan determinants.

Overall, we find evidence that a large fraction of microRNA promoters (and, thus, perhaps all promoters) are engaged by transcriptional programs that reflect future lifespan. We find little evidence that these programs are specific to certain tissue types; indeed it appears that at least three GFPs redundantly report on a single, organism-wide, cell non-autonomous, *daf-16*-independent state that is associated with future lifespan. The fact that half of the *promoter*::GFP strains we tested did *not* correlate with lifespan suggests that this global state is transcriptional in nature: any lifespan-associated post-transcriptional mechanism would affect *all* GFP transcripts or proteins identically, regardless of the promoter sequence driving GFP expression. This stands in contrast to the case of interindividual variation in *Phsp-16.2*::GFP after heat shock, which is related, via IIS activity, to a global state of enhanced or decreased protein stability (*Burnaevskiy et al., 2019*). At a minimum, the transcriptional state we identified reflects early changes in organismal physiology leading to early vs. late death; at most, this state may in fact determine those physiological changes.

## Results

### Longitudinal observation of PmiRNA::GFP reporters

To identify genes whose expression early in life is predictive of future lifespan, we selected 22 integrated miRNA fluorescent reporters (*PmiRNA*::GFP) from a larger library of 73 transgenic strains (*Martinez et al., 2008*), using as the only selection criterion the ability to detect reporter fluorescence with short (<100 ms) exposure times at ×5 magnification on our imaging system. Most of the miRNAs corresponding to the selected reporters have no reported function, and in particular, no lifespan phenotype upon knockout (*Table 1*).

Each reporter was crossed into the temperature-sensitive sterile strain *spe-9(hc88)* and examined in a high-density single animal culture device previously developed by our lab (*Figure 1a*; *Zhang et al., 2016*; *Pittman et al., 2017*) maintained at 25°C. For each *PmiRNA*::GFP;*spe-9(hc88)* strain, we collected bright-field and fluorescence images of each individual every 4 hr from hatch until death. Using an in-house image analysis pipeline, the *C. elegans* in each bright-field image was automatically identified, defining a mask that separates 'worm pixels' from the image background. We converted images of GFP fluorescence, which contain many thousands of pixel intensities within the image region corresponding to a single animal, into a single summary statistic. This allowed us to quantify *PmiRNA*::GFP expression over time within a single individual (*Figure 1b*, left; see

**Table 1.** miRNAs corresponding to *PmiRNA*::GFP reporters selected for this study and the predominant expression pattern and time window of expression, as observed at ×5 magnification.

Published regulatory functions and lifespan phenotypes for each miRNA are noted. Bold text indicates miRNAs that we found to be predictive of lifespan in this current study, as measured by correlation of *PmiRNA*::GFP expression with lifespan.

| miRNA | Expression pattern | Time window of expression | Regulatory function | Lifespan phenotype |
|---|---|---|---|---|
| *let-7* | Ubiquitous | Embryo–death | Developmental timing (*Reinhart et al., 2000*) | – |
| *lin-4* | Ubiquitous | L1–death | Developmental timing (*Wightman et al., 1993*) | *lin-4(e912)* are short-lived; *lin-4* overexpression extends lifespan (*Boehm, 2005*) |
| *mir-1* | Pharynx | Embryo–death | Synaptic function (*Simon et al., 2008*) | – |
| *mir-228* | Neurons | Embryo–death | – | *mir-228(n4382)* are long-lived; *mir-228* overexpression shortens lifespan (*Smith-Vikos et al., 2014*) |
| *mir-240–786* | Uterus, gonad sheath | L3–death | Defecation (*Kemp et al., 2012*) | – |
| *mir-241* | Hypodermis | L1–four dph | Developmental timing (*Abbott, 2005*) | – |
| *mir-242* | Neurons | Embryo–death | – | – |
| *mir-243* | Intestine | Embryo–death | – | – |
| *mir-246* | Gonad sheath | L4–death | – | *mir-246(n4636)* are short-lived; *mir-246* overexpression extends lifespan (*de Lencastre et al., 2010*) |
| *mir-360* | Pharynx | Embryo–death | – | – |
| *mir-47* | Hypodermis, vulva | Embryo–death | – | – |
| *mir-51* | Intestine | Embryo–death | Developmental timing (*Brenner et al., 2012*), GABAergic synapses (*Zhang et al., 2018*) | – |
| *mir-59* | Vulva | L4–death | – | – |
| *mir-60* | Intestine | Embryo–death | Oxidative stress response (*Kato et al., 2016*) | *mir-60(n4947)* are long-lived under oxidative stress conditions (*Kato et al., 2016*) |
| *mir-63* | Intestine | Embryo–death | – | – |
| *mir-788* | Hypodermis | Embryo–four dph | – | – |
| *mir-79* | Hypodermis | Embryo–five dph | Proteoglycan homeostasis (*Pedersen et al., 2013*) | – |
| *mir-793* | Neurons | Embryo–death | – | – |
| *mir-794* | Body wall muscle, intestine | Embryo–four dph | – | – |
| *mir-84* | Pharynx, vulva | L1–death | Developmental timing (*Abbott, 2005*) | – |
| *mir-85* | Gonadal sheath, uterus, spermatheca | L2–death | – | – |
| *mir-90* | Body wall muscle, vulva | Embryo–death | – | – |

Materials and methods for details). We then averaged the fluorescence measurements across all individuals to capture the overall population trend in expression of each reporter over time (*Figure 1b*, right, and 1 c), which is consistent with individual expression trends (*Figure 1—figure supplement 2*). For some reporters, such as *Pmir-788*::GFP, expression analysis was limited to shortened time windows, as late in life, reporter fluorescence becomes indistinguishable from background autofluorescence (see *Table 1*).

We observed several categories of temporal dynamics among the reporters. Expression of some, such as *Plet-7*::GFP, *Pmir-788*::GFP, *Pmir-79*::GFP, *Pmir-85*::GFP, and *Pmir-84*::GFP, peak during young adulthood and sharply decline thereafter, in some cases becoming undetectable with age. Expression of others, like *Pmir-793*::GFP, *Pmir-246*::GFP, and *Pmir-228*::GFP, also decrease with age but exhibit a slower decline or plateau after young adulthood. Overall, the majority of *PmiRNA*::GFP reporters decreased with age, in agreement with previously published microarray (*Ibáñez-*

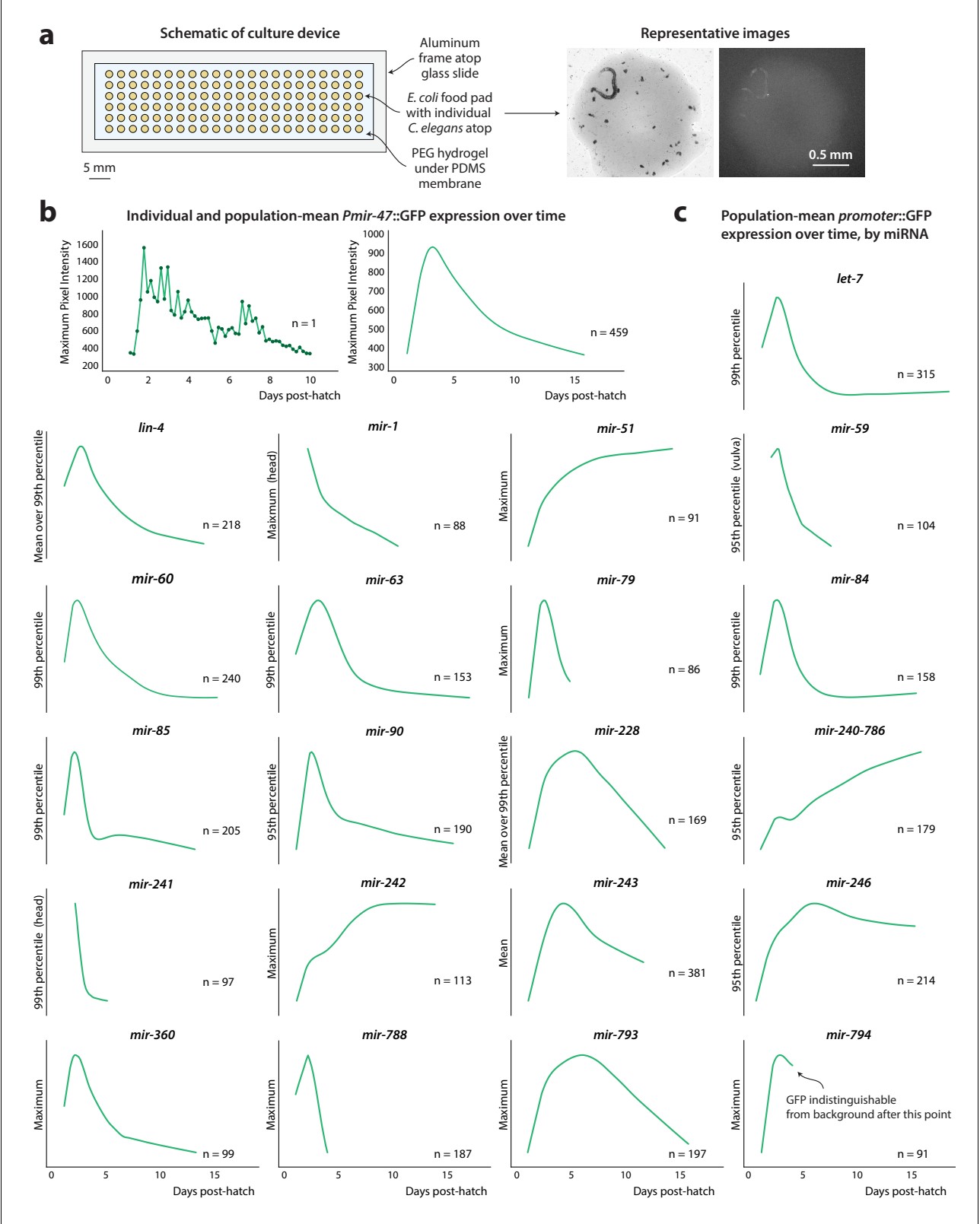

**Figure 1.** *PmiRNA*::GFP expression over time. (**a**) Schematic of high-density single animal culture device and representative bright-field and fluorescent images acquired from a single *Pmir-47*::GFP; *spe-9(hc88)* individual. (**b**) Timecourse of *Pmir-47*::GFP expression for a single animal from hatch until death (*left*). Expression is measured as the maximum pixel intensity within the image region comprising that individual, from images acquired every 4 hr. LOWESS regression showing the average population timecourse of *Pmir-47*::GFP expression, collated from five biological replicates comprising

*Figure 1 continued on next page*

*Figure 1 continued*

more than 400 individuals (*right*). (c) Population timecourse of expression for all *PmiRNA*::GFP reporters. (Fluorescent values are in arbitrary units; in all cases the bottom of the y-axis is approximately the noise floor of the camera sensor).

The online version of this article includes the following figure supplement(s) for figure 1:

**Figure supplement 1.** Representative fluorescence images taken of individuals at 3 dph (top right panels) and 7 dph (bottom right panels) from each *PmiRNA*::GFP reporter strain.

**Figure supplement 2.** Timecourse of expression for 10 randomly selected individuals from all *PmiRNA*::GFP reporters.

**Figure supplement 3.** LOWESS trends for expression of predictive *PmiRNA*::GFP reporters over time (solid line) and standard deviation (dashed line).

*Ventoso et al., 2006*) and small-RNA sequencing studies (*de Lencastre et al., 2010*; *Kato et al., 2011*). Unusual for miRNAs, *Pmir-51*::GFP, *Pmir-242*::GFP, and *Pmir-240–786*::GFP show an increase in expression over time (*Figure 1c*). This increase was also observed with small RNA sequencing (*de Lencastre et al., 2010*; *Kato et al., 2011*), indicating that the *PmiRNA*::GFP reporters generally reflect age-related trends in endogenous transcription of miRNAs. In contrast to sequencing, however, this work provides a more detailed picture of expression trends over time.

## PmiRNA::GFP reporters are predictive biomarkers of future lifespan

In order to visualize the relationship between *PmiRNA*::GFP expression and longevity (if any), we binned populations of the *PmiRNA*::GFP reporter strains into cohorts based on eventual lifespan (*Figure 2a*) and plotted average reporter expression for each lifespan-cohort over time (*Figure 2b and d*). For a subset of *PmiRNA*::GFP reporters, the average expression level and rate of change of expression appeared to differ substantially between cohorts in mid- to late- adulthood, indicating that reporter expression may be predictive of an individual's future lifespan. To test this quantitatively, we performed a multivariate regression on each individual animal's average level of GFP expression and the trend in GFP expression (positive or negative slope) against future lifespan (*Figure 2c and d*; *Table 2*). Because the temporal dynamics of the different *PmiRNA*::GFP reporters vary substantially, we used a sliding time window to determine the optimal span of expression data to include in the regression for each reporter strain. We examined GFP expression bounded between 3 days post-hatch (dph; approximately the first day of adulthood) and the time of 90% population survival. Note that that the time of 90% survival varied among strains, due to per-strain differences and, to a lesser extent, batch and seasonal effects (*Table 2*). We performed regressions using GFP levels at all possible windows within the overall bounds, by independently moving beginning and ending timepoints of the window at 12 hr intervals. Individuals that died before or within the chosen window were censored from analysis to avoid truncation effects (these short-lived individuals necessarily have fewer expression measurements, which can confounding analysis). The window that resulted in the maximum correlation between expression and lifespan was selected (*Table 2*). Similar results obtain using a simple, fixed time-window between 3 days post-hatch and the 90% survival timepoint, or repeating the variable-window analysis bounded between three dph and the 95% survival timepoint (tables S1 and S2 in *Supplementary file 1*).

Expression of 10 of the 22 tested *PmiRNA*::GFP reporters exhibited an ability to reproducibly predict lifespan with a joint correlation coefficient ($R^2$) of at least 0.15: the reporters for miRNAs *lin-4*, *mir-47*, *mir-60*, *mir-85*, *mir-90*, *mir-228*, *mir-240–786*, *mir-243*, *mir-246*, and *mir-793* (*Table 2*). The $R^2$ value in this case represents the fraction of total inter-individual variation in lifespan that can be accounted for by the inter-individual variation in measured GFP levels of a single reporter.

The ability to predict lifespan does not appear to be a generic property of promoter::GFP constructs, as over half of the examined reporters did not correlate substantially with lifespan. Furthermore, two non-miRNA reporters we tested, *Pmyo-2*::GFP and *Pcpna-2*::GFP, were not substantially predictive of lifespan (*Figure 2—figure supplement 1* and table S3 in *Supplementary file 1*). Moreover, the optimal time window for regression (i.e. the time window of expression data producing the maximum correlation coefficient value) varied between reporters, suggesting that the reporters may provide readouts of different phases of the aging process. Some reporters exhibited broad (at least 2 days) optimal time windows spanning young adulthood (*Pmir-243*::GFP, *Pmir-85*::GFP) or mid-to-late adulthood (*Plin-4*::GFP, *Pmir-47*::GFP, *Pmir-228*::GFP, *Pmir-246*::GFP, *Pmir-793*::GFP). Other reporters like *Pmir-60*::GFP, *Pmir-90*::GFP, and *Pmir-240–786*::GFP were optimally predictive in

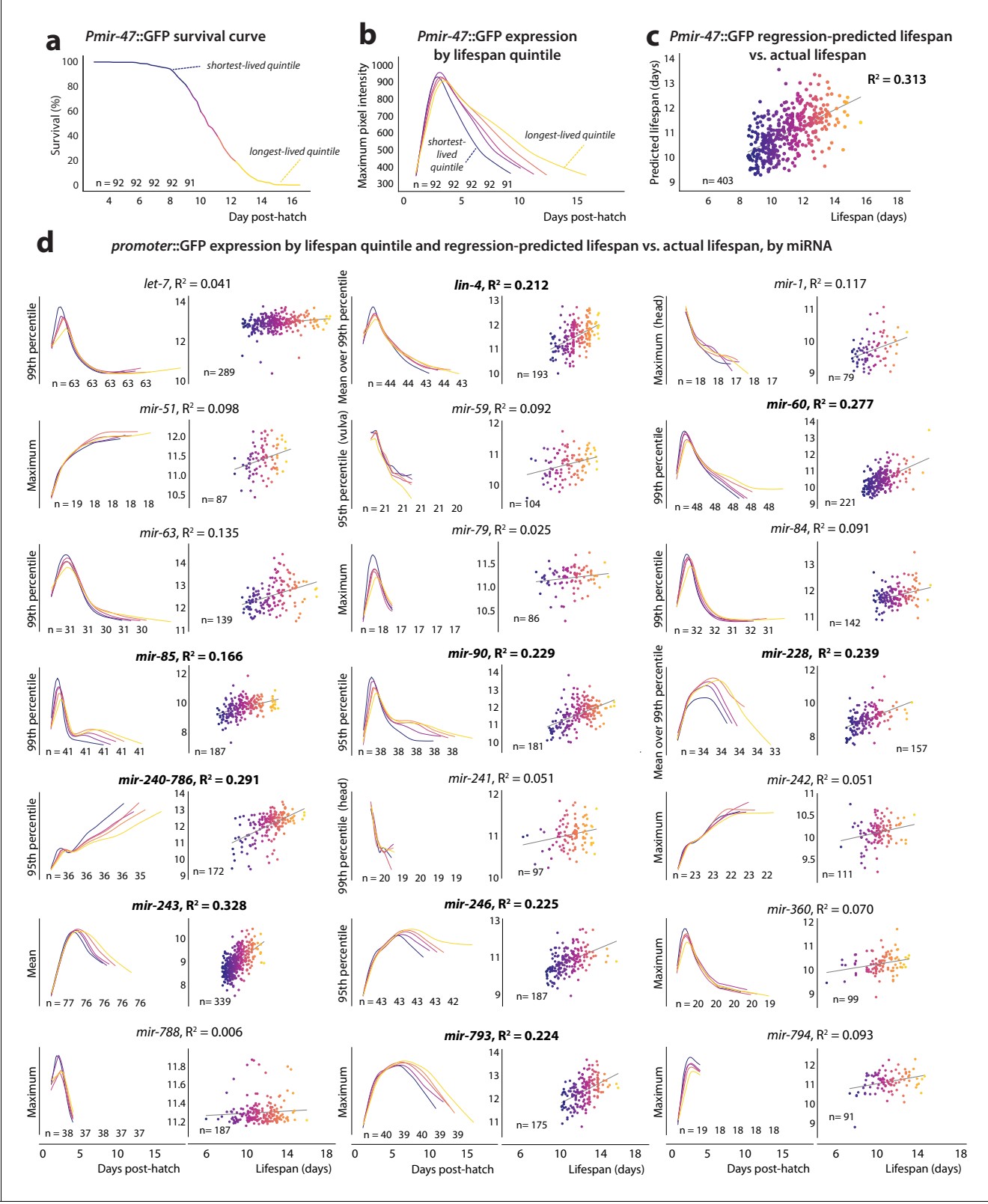

**Figure 2.** *PmiRNA*::GFP expression vs. future lifespan. (**a**) Survival curve for *Pmir-47*::GFP; *spe-9(hc88)* individuals pooled from five biological replicates. Animals are binned into color-coded quintiles based on eventual lifespan. The number of individuals in each quintile is indicated. (**b**) LOWESS regression of each quintile's average *Pmir-47*::GFP expression (as measured by maximum intensity per individual at each timepoint) is plotted over time for the quintiles in panel a. (**c**) Joint regression of both the mean level of each individual's *Pmir-47*::GFP expression between 5 and 8.5 days post-hatch,

*Figure 2 continued on next page*

*Figure 2 continued*

and the slope of that expression over that time, against future lifespan yields an $R^2$ of 0.313. Each dot represents an individual animal and is color-coded based on lifespan. (d) Cohort-level expression timecourse and joint regression of slope and mean expression against future lifespan for all *PmiRNA*::GFP reporters. This mean and slope were measured across the optimal time window for each reporter's correlation with future lifespan (specified in *Table 2*). Correlation coefficients > 0.15 are indicated by bolded text.

The online version of this article includes the following figure supplement(s) for figure 2:

**Figure supplement 1.** Expression of non-miRNA reporters *Pcpna-2*::GFP and *Pmyo-2*::GFP is plotted over time for each lifespan cohort.

narrower time windows centered at late adulthood. For the latter set of reporters, the average expression value tended to be more predictive of future lifespan than the slope of an individual's expression trendline. For the earlier predictors, the slope of the trendline tended to be as or more predictive of lifespan than the the average expression value (*Table 2*). Neither the maximum expression value an individual achieved nor the timepoint at which maximum expression was reached were substantially predictive of future lifespan for any of the *PmiRNA*::GFP reporters (table S4 in *Supplementary file 1*). A positive correlation between *Pmir-246*::GFP expression and lifespan has been previously published and *lin-4* and *mir-228* are known to play roles in longevity and the aging

**Table 2.** Correlation of *PmiRNA*::GFP reporters with lifespan.

A joint regression of slope and average expression against lifespan was performed using a sliding time window (minimum width of 12 hr) beginning at 3 days post-hatch and ending at the 90% survival timepoint, which we observed to be variable between strains. The optimal time window and highest correlation achieved is reported. $R^2$ values exceeding 0.15 are indicated by bolded text. The 95% confidence interval (CI) for each joint correlation coefficient is also shown. The direction of correlation, derived from the individual regression on slope and mean expression (which we observed to always correlate in the same direction), is indicated by (+) or (-), respectively.

| miRNA | N | Measure of pixel intensity | Time window (dph) | Slope $R^2$ | Mean $R^2$ | Joint $R^2$ | Joint 95% CI |
|---|---|---|---|---|---|---|---|
| *let-7* | 289 | 99th percentile | 6.0–9.5 | 0.024 | 0.012 | 0.041 | [−0.002, 0.085] |
| *lin-4* | 193 | Mean over 99th percentile | 4.5–9.0 | 0.211 | 0.029 | **0.212 (+)** | [0.124, 0.306] |
| *mir-1* | 79 | Maximum (head) | 6.0–7.5 | 0.102 | 0.025 | 0.117 | [0.008, 0.234] |
| *mir-47* | 403 | Maximum | 5.0–8.5 | 0.158 | 0.158 | **0.313 (+)** | [0.240, 0.388] |
| *mir-51* | 87 | Maximum | 8.0–8.5 | 0.025 | 0.048 | 0.098 | [−0.010, 0.211] |
| *mir-59* | 104 | 95th percentile (vulva) | 4.5–5.0 | 0.002 | 0.092 | 0.092 | [−0.020, 0.186] |
| *mir-60* | 221 | 99th percentile | 7.5–8.5 | 0.035 | 0.238 | **0.277 (+)** | [0.137, 0.402] |
| *mir-63* | 139 | 99th percentile | 5.0–9.5 | 0.118 | 0.053 | 0.135 | [0.047, 0.230] |
| *mir-79* | 86 | Maximum | 3.5–4.0 | 0.005 | 0.010 | 0.025 | [−0.033, 0.076] |
| *mir-84* | 142 | 99th percentile | 9.0–9.5 | 0.000 | 0.091 | 0.091 | [−0.016, 0.187] |
| *mir-85* | 187 | 99th percentile | 3.5–7.0 | 0.157 | 0.052 | **0.166 (+)** | [0.075, 0.257] |
| *mir-90* | 181 | 95th percentile | 8.0–8.5 | 0.014 | 0.215 | **0.229 (+)** | [0.116, 0.340] |
| *mir-228* | 157 | Mean over 99th percentile | 5.0–7.0 | 0.194 | 0.070 | **0.239 (+)** | [0.129, 0.350] |
| *mir-240–786* | 172 | 95th percentile | 7.5–8.5 | 0.043 | 0.268 | **0.291 (-)** | [0.168, 0.419] |
| *mir-241* | 97 | 99th percentile (head) | 3.0–3.5 | 0.013 | 0.037 | 0.051 | [−0.038, 0.133] |
| *mir-242* | 111 | Maximum | 4.5–5.5 | 0.050 | 0.010 | 0.051 | [−0.041, 0.130] |
| *mir-243* | 339 | Mean | 3.5–7.5 | 0.299 | 0.110 | **0.328 (+)** | [0.254, 0.409] |
| *mir-246* | 187 | 95th percentile | 6.5–8.5 | 0.124 | 0.124 | **0.225 (+)** | [0.124, 0.332] |
| *mir-360* | 99 | Maximum | 4.0–5.0 | 0.070 | 0.000 | 0.070 | [−0.032, 0.157] |
| *mir-788* | 187 | Maximum | 3.0–3.5 | 0.000 | 0.005 | 0.006 | [−0.029, 0.037] |
| *mir-793* | 175 | Maximum | 6.0–10.0 | 0.166 | 0.091 | **0.224 (+)** | [0.146, 0.312] |
| *mir-794* | 91 | Maximum | 3.0–3.5 | 0.009 | 0.082 | 0.093 | [−0.032, 0.209] |

process; however, to our knowledge, an association between lifespan and expression of the other miRNAs in unperturbed *C. elegans* has not been reported (*Table 1*).

In each case, extended lifespan was correlated with retention of young-adult gene-expression levels and/or trends. Specifically, all reporters except for *Pmir-240–786*::GFP both decreased in expression throughout aging and correlated positively with future lifespan. Conversely, *Pmir-240–786*::GFP increases in expression with age (*Figure 1*) and was negatively correlated with future lifespan. In other words, for reporters where expression levels start high and decrease over time, long lifespan is predicted by high ('youthful') expression levels and a delay in their decline. By the same token, for *Pmir-240–786*::GFP, with low young-adult expression that increases with age, extended periods of 'youthful' low expression predict extended lifespan.

We next examined how the lifespan-predictive power of the *PmiRNA*::GFP reporters changes throughout adulthood. For every 24 hr period post-hatch, we jointly regressed each individual's average expression and the slope of the expression trend over those 24 hr against that individual's eventual lifespan (*Figure 3a*). We plotted the resulting correlation coefficient over time to illustrate how the predictive power of the reporter changes with aging (*Figure 3a and b*). Expression of predictive *PmiRNA*::GFP reporters generally begins to correlate with future lifespan when measured at mid to late adulthood, becoming more predictive over time. Comparing the correlation plot with the population survival curve revealed that reporter predictivity peaks simultaneously with the beginning of population die-off, continuing to significantly correlate with lifespan as more individuals in the population perish (*Figure 3a and b*). However, reporters do not simply correlate with imminent death, as for this analysis we excluded individuals that perished within the 24 hr time window. Moreover, excluding even those that perish within the subsequent 24 or 48 hr does not abolish predictive power for most reporters (*Figure 3—figure supplement 1*). None of the reporters we examined showed substantial correlation with future lifespan at timepoints less than 5 days post-hatch (roughly young adulthood). This result would be expected if reporter expression levels were identical across individuals early in life, and individuals' expression levels only diverged later in adulthood. However, we found that the degree of inter-individual variability in reporter levels is largely consistent throughout life (*Figure 1—figure supplements 2* and *3*). This suggests that only at mid-adulthood do pre-existing inter-individual differences in *PmiRNA*::GFP expression become coupled to the aging process and future lifespan.

Interestingly, we found that autofluorescence intensity, a phenomenological measure that negatively correlates with lifespan (*Pincus et al., 2016*), is predictive of future longevity at approximately the same time in adulthood as are most of the *PmiRNA*::GFP reporters we examined (*Figure 3—figure supplement 2*). However, *PmiRNA*::GFP expression and autofluorescence provide largely non-redundant information with respect to future lifespan, based on the near-additivity of correlation coefficients in single vs. joint regression (table S5 in *Supplementary file 1*). This suggests that although both act as biomarkers of future lifespan, they likely report on distinct aspects of the aging process.

## Variable expression of mir-47 and mir-243 are not determinants of individual variability in lifespan

Expression of the *PmiRNA*::GFP constructs can only correlate with future lifespan if there is some biological process which both (a) determines individual lifespan and (b) directly or indirectly regulates transcription of the GFP reporters. To learn more about the nature of this process, we investigated the genetic requirements for the correlation between lifespan and GFP levels in the two most lifespan-predictive reporters, *Pmir-47*::GFP and *Pmir-243*::GFP.

One straightforward possibility is that the *PmiRNA*::GFP constructs are honest reporters for their endogenous, cognate miRNAs, which determine individual lifespan by variability in their own expression ('Direct Causation' model; *Figure 4a*). Alternately, the endogenous miRNAs themselves may have nothing to do with the true lifespan-determining pathway; in this scenario, reporter expression is a downstream readout of pathways, processes and/or transcription factor activity that affect lifespan but do not require the activity of the endogenous miRNA ('Bystander Correlation' model; *Figure 4a*). Under the former model, activity of the cognate miRNA is a critical causal link to lifespan; absent that miRNA, variability in GFP would be decoupled from variability in lifespan. Under the latter model, loss of the miRNA would not influence the relationship between GFP expression and lifespan.

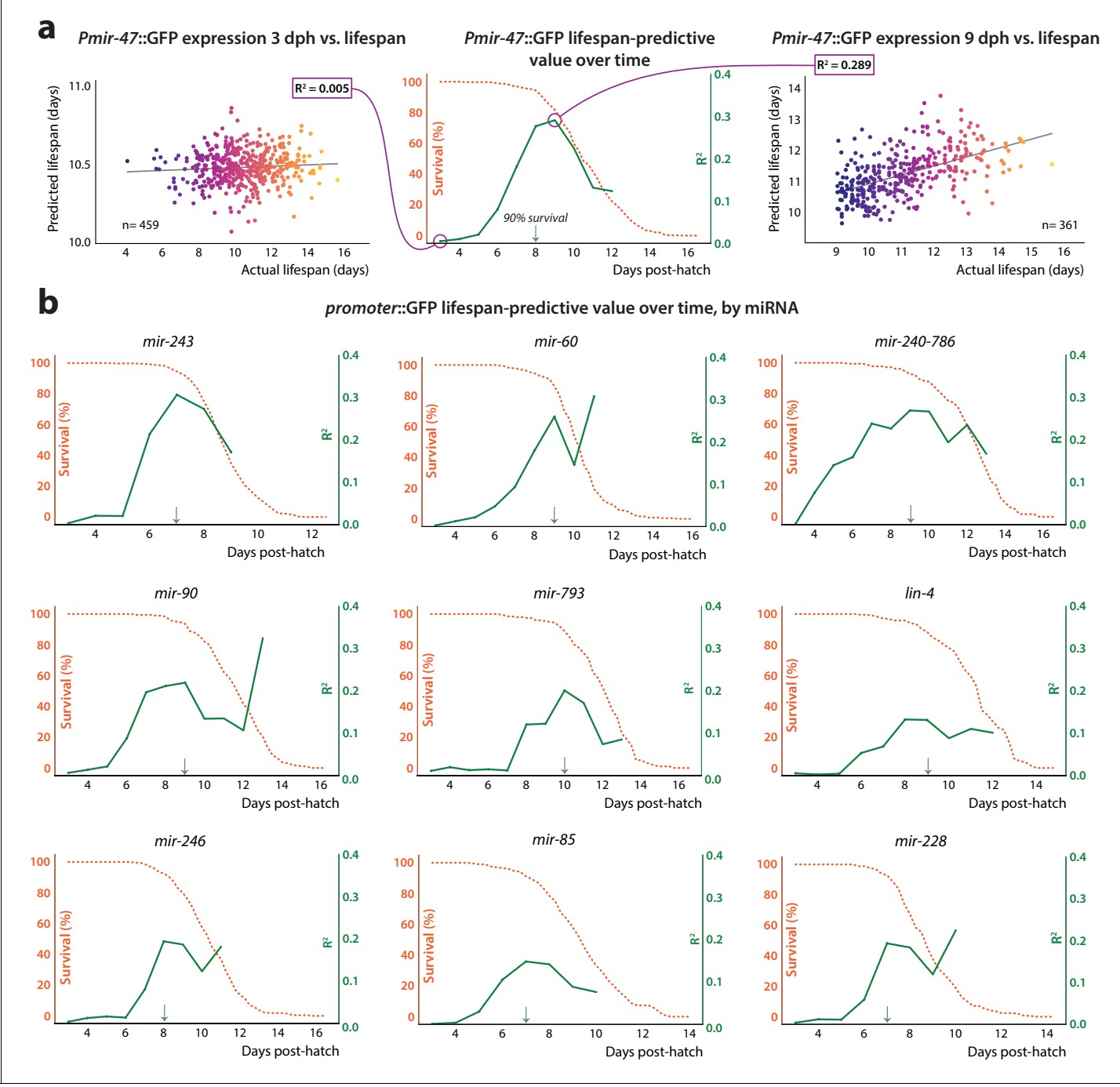

**Figure 3.** Lifespan-predictive ability vs.time. (a) Construction of correlation plot for *Pmir-47*::GFP. Mean and slope of expression values for each individual during a sliding 24 hr time window, starting at 3 dph, were regressed against future lifespan. The regressions for 48–72 hr (window centered at 3 dph, *left*) and 192–216 hr (9 dph, *right*) are shown. At center, the correlation coefficient from each regression is plotted against the age at the middle of the regression window (solid line). The survival curve for the population is overlaid (dashed line). The 90% survival timepoint is indicated by the green arrow. (b) Correlation plots for remainder of predictive *PmiRNA*::GFP reporters.

The online version of this article includes the following figure supplement(s) for figure 3:

**Figure supplement 1.** Mean and slope of PmiRNA::GFP expression in a sliding 24 hr window was regressed against future lifespan, only for individuals surviving the subsequent 24 hr (green), 48 hr (purple), and 72 hr (orange).

**Figure supplement 2.** Mean and slope of *PmiRNA*::GFP expression (green) and the 95th percentile of autofluorescence intensity (purple) for each individual during a 24 hr time window were regressed on future lifespan.

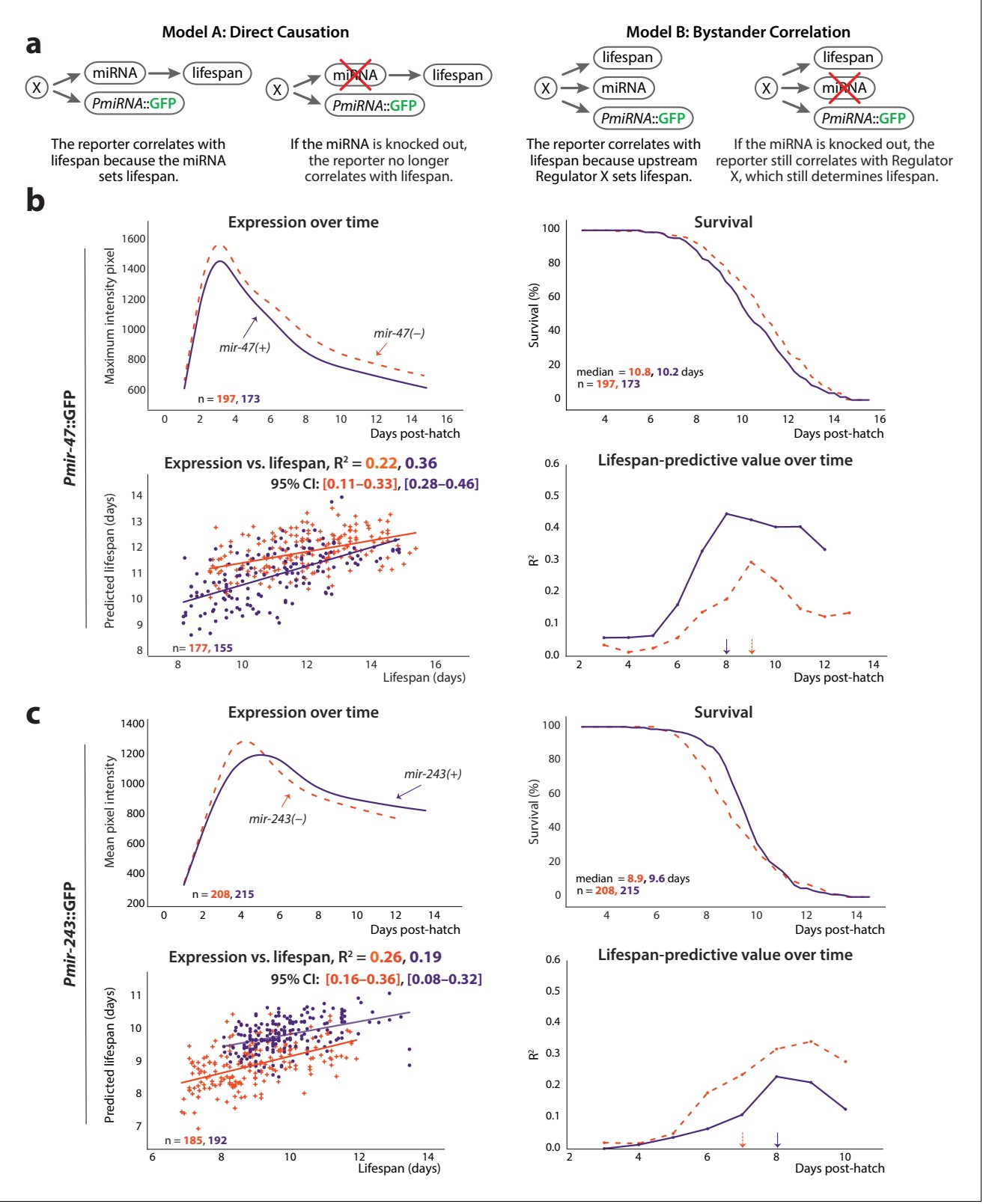

**Figure 4.** Non-involvement of endogenous miRNAs. (a) Two models of the possible relationship between the endogenous miRNA and lifespan: 'direct causation' in which the miRNA is directly involved in determining individual lifespans, and 'bystander correlation' in which it is not. The predicted effect of knocking out the endogenous miRNA on the correlation between *PmiRNA*::GFP levels and future lifespan is shown for each scenario. (b) Comparison of *Pmir-47*::GFP expression over time (*top left*), survival (*top right*), regression on slope and mean of expression against lifespan (*bottom right*), and

*Figure 4 continued on next page*

*Figure 4 continued*

lifespan-predictive value (*bottom left*) between *mir-47(+)* (solid lines) and *mir-47(gk167)* (dashed lines) backgrounds. The time windows for regression were those that maximized correlation with future lifespan: 72–194 and 72–215 hr post-hatch for *mir-47(+)* and *mir-47(gk167)*, respectively. (**c**) Comparison of *Pmir-243*::GFP expression over time (*top left*), survival (*top right*), regression on slope and mean of expression against lifespan (*bottom left*), and lifespan-predictive value (*bottom right*) between *mir-243(+)* (solid lines) and *mir-243(n4759)* (dashed lines) backgrounds. The time windows for regression were 72–192 and 72–164 hr post-hatch for *mir-243(+)* and *mir-243(n4759)*, respectively. The 90% survival ages for each genotype are indicated by arrows. The 95% CI for each regression correlation coefficient is shown in brackets. In scatter plots, the (o) and (+) symbols correspond to the regressions in wild-type or miRNA-mutant genotypes, respectively. All figures pool data from two biological replicates.

To distinguish these two possibilities, we tested whether *Pmir-47*::GFP and *Pmir-243*::GFP were still predictive of lifespan in a genetic background lacking the cognate endogenous miRNAs (*mir-47 (gk167)* and *mir-243(n4759)*, respectively). As before, we performed regressions on average expression and the slope of the expression trendline in windows between three dph and the 90% survival timepoint; we also measured the lifespan-predictive value of the reporters over time in discrete 24 hr time periods. The correlation of *Pmir-47*::GFP expression with lifespan was somewhat reduced in the *mir-47(gk167)* mutant compared to *mir-47(+)* ($R^2$ = 0.22 vs. 0.36) and exhibited a slightly different temporal window for lifespan prediction (7–9 dph vs. 6–8 dph) (*Figure 4b*). The correlation of *Pmir-243*::GFP expression with lifespan was actually increased in the *mir-243(n4759)* mutant compared to *mir-243(+)* ($R^2$ = 0.257 vs. 0.193), and exhibited a different temporal window (6–7 dph vs. 7–8 dph) (*Figure 4c*). Overall lifespans were unaffected in *mir-47(gk167)* and *mir-243(n4759)* compared to strains wild-type at those loci (median lifespan = 10.8 vs. 10.2 days and 8.9 vs. 9.6 days, respectively). Because both reporters correlated positively with lifespan, a short-lived phenotype might be expected if either miR-243 or miR-47 played a functional role in determining lifespan. The lack of substantial lifespan difference further suggests that neither miRNA plays a direct functional role in determining lifespan.

Overall, the correlation of *Pmir-47*::GFP and *Pmir-243*::GFP with lifespan was not abolished in the absence of the endogenous miRNAs. This provides evidence for the 'Bystander Correlation' model of *Figure 4a*, implying that lifespan may be largely determined by transcriptional regulators of the *PmiRNA*::GFP transgenes (and presumably of the endogenous miRNAs as well), or regulatory pathways/processes even further upstream, rather than via the activity of the miRNAs themselves.

## Lifespan-predictive abilities of Pmir-47::GFP and Pmir-243::GFP are independent of daf-16

As the abundance and/or activity of miR-47 and miR-243 does not appear to directly affect lifespan, *Pmir-47*::GFP and *Pmir-243*::GFP must instead serve as markers of other pathways or processes that influence lifespan. We wondered if these might include insulin/insulin-like growth factor (IGF-1) signaling (IIS), a canonical aging pathway in *C. elegans* that is highly conserved across taxa (*Kenyon, 2010*). The primary effector of the IIS pathway in *C. elegans* is the FOXO transcription factor DAF-16, which modulates the expression of a number of downstream genes that influence aging and lifespan (*Kenyon, 2010*; *Murphy et al., 2003*). DAF-16 activity is a common genetic requirement for genes, pathways, and processes that have been reported to increase lifespan in *C. elegans*, including several microRNAs (*Pincus et al., 2011*; *Boehm, 2005*; *Boulias and Horvitz, 2012*). It is therefore plausible that inter-individual variability in insulin signaling or signal-responsiveness leads to variability in DAF-16 activity and thus to inter-individual differences in subsequent lifespan. Indeed, two previously characterized lifespan-predictive gene-expression reporters, *Psod-3*::GFP and *Pmir-71*::GFP, both require functional DAF-16 to correlate with future lifespan. This suggests those transgenes predict future lifespan by reporting on inter-individual variability in DAF-16 (and presumably IIS) activity. To test if *Pmir-47*::GFP and *Pmir-243*::GFP similarly report on DAF-16 activity, we examined whether GFP levels still correlated with lifespan in the absence of DAF-16 (*Figure 5a*). Specifically, we crossed each reporter into a *daf-16(mu86);spe-9(hc88)* background and assayed lifespan-predictive ability in side-by-side experiments with the reporter in the reference background (*spe-9(hc88)* only).

Regression on average expression and the slope of the expression trendline between 3 dph and the 90% survival timepoint showed that the correlation of *Pmir-47*::GFP or *Pmir-243*::GFP expression with lifespan was not substantially suppressed in the absence of *daf-16* (bottom left panels of

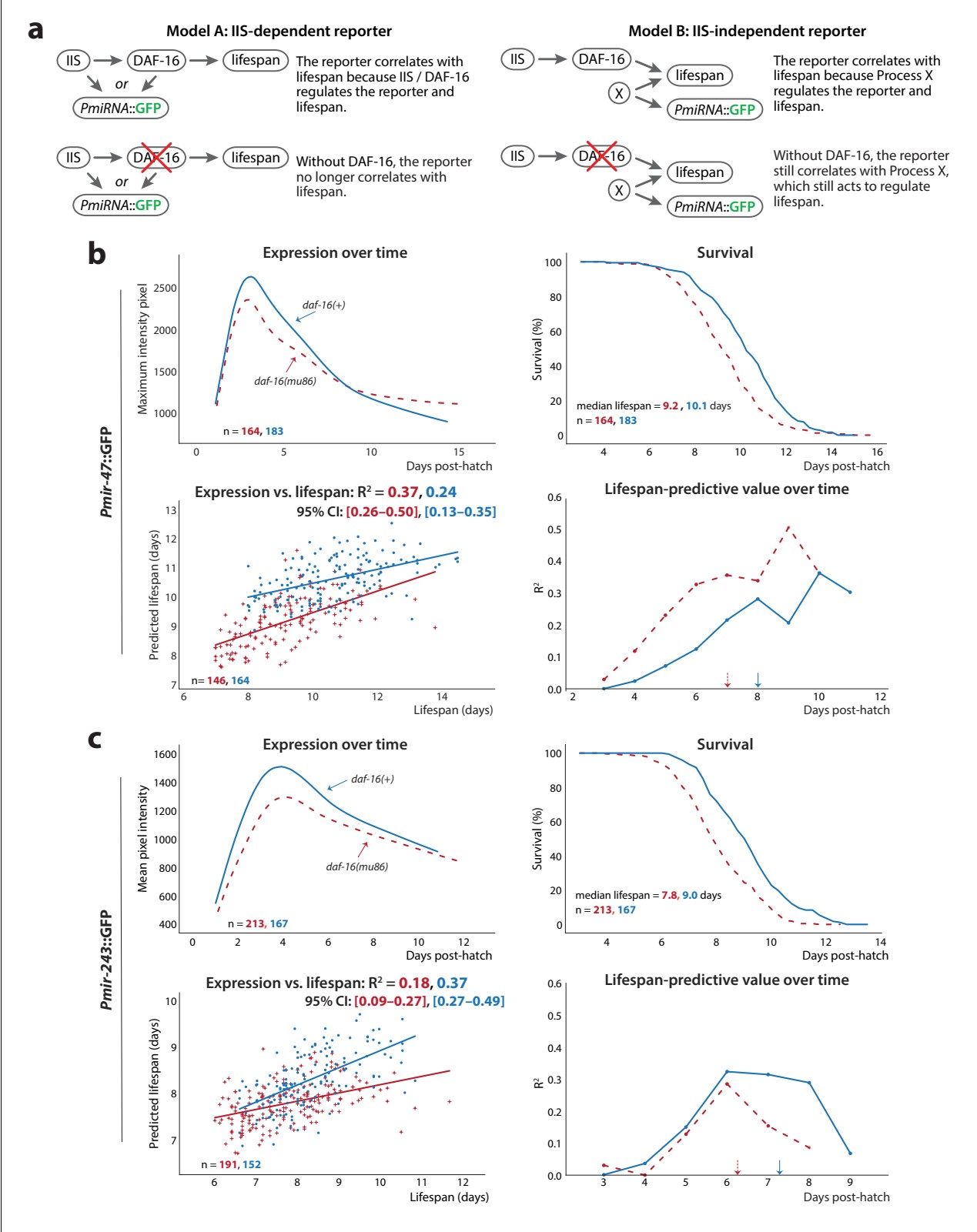

**Figure 5.** Non-involvement of IIS. (**a**) For *PmiRNA*::GFPs to correlate with lifespan, each must report on some lifespan-determining process. The reported-on process may be IIS-dependent (*left*) or IIS-independent (*right*); these scenarios can be distinguished by the effects of knocking out the IIS effector *daf-16* on the correlation between *PmiRNA*::GFP and future lifespan. (**b**) Comparison of *Pmir-47*::GFP expression over time (*top left*), survival (*top right*), regression on slope and mean of expression against lifespan (*bottom right*), and lifespan-predictive value (*bottom left*) between *daf-16(+)*

Figure 5 continued

(solid lines) and *daf-16(mu86)* (dashed lines) backgrounds. The optimal time windows for regression were 72–190 and 72–164 hr post-hatch for *daf-16* wild-type and *daf-16(mu86)*, respectively. (c) Comparison of *Pmir-243*::GFP expression over time (*top left*), survival (*top right*), regression on slope and mean of expression against lifespan (*bottom left*), and lifespan-predictive value (*bottom right*) between *daf-16(+)* (solid lines) and *daf-16(mu86)* (dashed lines) backgrounds. The optimal time windows for regression were 72–158 and 72–144 hr post-hatch for *daf-16(+)* and *daf-16(mu86)*, respectively. The 90% survival ages for each genotype are indicated by arrows. The 95% CI for each regression correlation coefficient is shown in brackets. In scatter plots, the (o) and (+) symbols correspond to the regressions in *daf-16(+)* and *daf-16(mu86)*, respectively. All figures pool data from two biological replicates.

*Figure 5b and c*, respectively). For *Pmir-47*::GFP, the $R^2$ values vs. lifespan were 0.24 for *daf-16(+)* and 0.37 for *daf-16(-)*; for *Pmir-243*::GFP, the $R^2$ values were 0.37 for *daf-16(+)* and 0.18 for *daf-16(-)*. Similarly, plotting the lifespan-predictive value of *Pmir-47*::GFP or *Pmir-243*::GFP over time shows that the predictive power of each reporter is not reduced in the *daf-16(mu86)* background (bottom right of *Figure 5b and c*, respectively). As expected, the predictive windows are slightly shifted between *daf-16(mu86)* and wild-type backgrounds due to the short-lived phenotype of *daf-16(mu86)* (top right of *Figure 5b and c*). Interestingly, peak levels of *Pmir-243*::GFP and *Pmir-47*::GFP were slightly but consistently reduced in the *daf-16(mu86)* background, suggesting that DAF-16 may play a minor role in the transcriptional regulation of these reporters.

## Pmir-240–786::GFP, Pmir-793::GFP, and Pmir-47::GFP hierarchically report on a single lifespan-determining process

The lifespan-predictive *PmiRNA*::GFP reporters we identified cannot all provide independent information. Because the correlation coefficients with future lifespan (*Table 2*) sum to more than one, at least some subset of these reporters must necessarily be at least partially redundant with one another. This raises an obvious question: to what degree are the reporters independent? At one extreme, all 10 reporters might reflect the activity of a single lifespan-determining pathway. Alternately, there may be a small handful of such pathways that are reported on by different sets of the *PmiRNA*::GFPs.

If two reporters provide information about distinct, independent biological processes, then when both reporters are measured in the same individual, the lifespan estimate using both reporters will be better than when using either individually. More specifically, if the information that each reporter provides about future lifespan is completely independent, then the $R^2$ of a joint regression using both measurements to predict future lifespan will be the sum of the $R^2$-values from the two single regressions. Alternatively, if two *PmiRNA*::GFP transgenes both report on the same process, then expression of one transgene will be correlated with the other, and, equivalently, the joint regression will not be substantially more predictive of lifespan than either single regression alone (*Figure 6a*).

To begin to count the number of lifespan-determining processes reported on by the *PmiRNA*::GFPs we identified, we investigated the degree of redundancy among *Pmir-47*::GFP, *Pmir-793*::GFP and *Pmir-240–786*::GFP. We chose these reporters because they are among the strongest predictors of future lifespan, and can be easily spatially resolved because they are primarily expressed in distinct tissues. We thus constructed three dual-reporter strains comprising all pairs of these three transgenes, and for each strain measured GFP levels of both transgene independently. In specific, we manually annotated fluorescence images to delineate tissues in which each transgene was specifically expressed, and extracted our GFP measurements from those regions only, rather than the whole worm (see Materials and methods and *Figure 6b–d*).

To determine whether each *PmiRNA*::GFP in the dual-reporter strain provided independent information about longevity, we regressed each reporter individually against future lifespan and compared the resulting $R^2$-values to that produced by multivariate regression of both reporters against lifespan. As a complementary analysis, we also calculated the semipartial correlation coefficient between each single reporter and future lifespan, statistically controlling for the expression of the other reporter. Specifically, the $R^2$ from semipartial correlation between reporter A and lifespan, controlling for reporter B, represents the fraction of variation in lifespan that is uniquely associated with variability in reporter A but not reporter B. In the case that reporter A is fully redundant with reporter B, then reporter A will no longer correlate with future lifespan at all after controlling for the expression levels of reporter B (semipartial $R^2 = 0$).

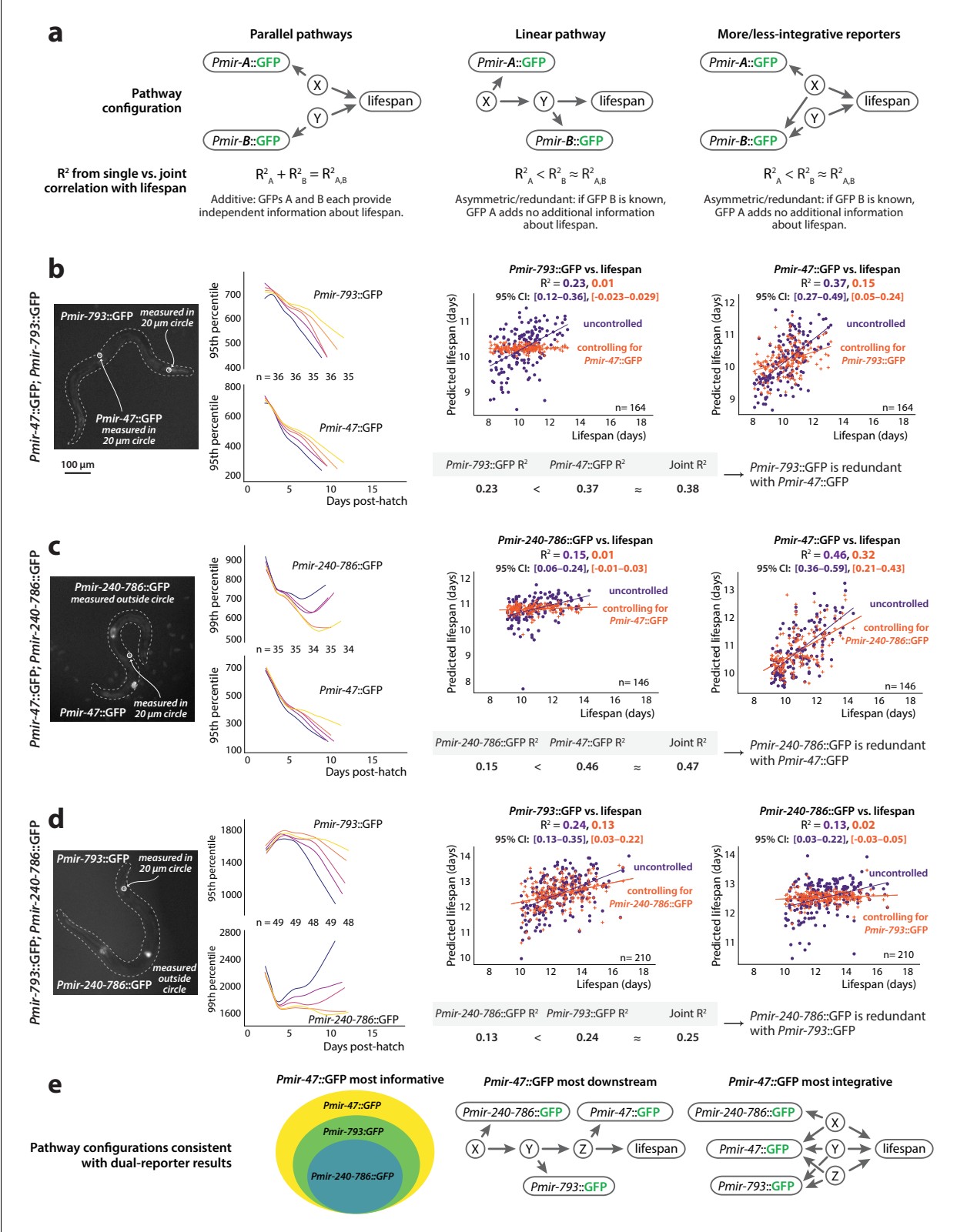

**Figure 6.** Redundancy of reporters. (**a**) Three models depicting possible additive or redundant relationships between lifespan-determining processes X and Y and two *PmiRNA*::GFP reporters A and B, with respect to their correlation vs. future lifespan. (**b**) *Pmir-793*::GFP and *Pmir-47*::GFP expression for different lifespan cohorts over time (*left*) and regression on slope and mean of each reporter from 3 to 8 dph controlled for expression of the other reporter (*right*). (**c**) *Pmir-47*::GFP and *Pmir-240–786*::GFP expression for different lifespan cohorts over time (*left*) and regression on slope and mean of

*Figure 6 continued on next page*

*Figure 6 continued*

each reporter from 3 to 9 dph controlled for expression of the other reporter (*right*). (d) *Pmir-793*::GFP and *Pmir-240–786*::GFP expression for different lifespan cohorts over time (*left*) and regression on slope and mean of each reporter from 3 to 10 dph controlled for expression of the other reporter (*right*). The tables in panels a–c compare correlation coefficients for single *PmiRNA*::GFP reporters to the joint correlation coefficient obtained by regressing on both reporters. In scatter plots, the (o) and (+) symbols correspond to the uncontrolled and controlled regressions, respectively. All figures represent data from two biological replicates. (e) Interpretations of the relationships among these three reporters and the reported-on lifespan-determining processes. Informational hierarchy (*left*): *Pmir-47*::GFP provides strictly more information about future lifespan than the other two reporters, and *Pmir-793*::GFP provides strictly more information than *Pmir-240–786*::GFP expression. This is consistent with (at least) two different mechanistic hierarchies. Signaling hierarchy (*center*): the informational hierarchy may suggest that *Pmir-47*::GFP provides lifespan-predictive information strictly more downstream than the other reporters. Integration hierarchy (*right*): alternately, *Pmir-47*::GFP could provide information that integrates among more independent lifespan determinants.

The online version of this article includes the following figure supplement(s) for figure 6:

**Figure supplement 1.** Pairwise correlations among GFP levels 3–8dph for each biomarker.

In the dual *Pmir-47*::GFP; *Pmir-793*::GFP strain, expression of each reporter is individually predictive of future lifespan, as we previously observed (***Figure 6b***, $R^2$ = 0.37 and 0.23 for *Pmir-47*::GFP and *Pmir-793*::GFP respectively). We found that *Pmir-793*::GFP expression correlates strongly with *Pmir-47*::GFP expression (***Figure 6—figure supplement 1***), which suggests that they are co-regulated and thus potentially redundant with respect to future lifespan. Indeed, joint regression incorporating both reporters produces a correlation coefficient no greater than regressing against *Pmir-47*:: GFP alone (***Figure 6b***, $R^2$ = 0.38). This suggests that *Pmir-793*::GFP levels provide no information about future lifespan that is not also reflected in *Pmir-47*::GFP expression. As a confirmation, the semipartial correlation between *Pmir-793*::GFP and lifespan, controlling for *Pmir-47*::GFP, is (as expected) nearly zero (***Figure 6b***, left plot). In contrast, *Pmir-47*::GFP expression remains somewhat predictive of lifespan even after controlling for *Pmir-793*::GFP. Thus, *Pmir-47*::GFP and *Pmir-793*:: GFP are not independent predictors of future lifespan and likely report on a single lifespan-determining pathway. Because *Pmir-47*::GFP expression contains strictly more information about lifespan than *Pmir-793*::GFP, we infer that *Pmir-47*::GFP likely reports on activity of that pathway at a somewhat more downstream position with respect to lifespan determination, or integrates additional information from a separate pathway (***Figure 6a***, center and right).

Similar results obtain for *Pmir-47*::GFP and *Pmir-240–786*::GFP. Expression of each reporter was individually predictive of lifespan (***Figure 6c***, $R^2$ = 0.46 and 0.15 for *Pmir-47*::GFP and *Pmir-240–786*::GFP, respectively), but expression levels of the reporters are correlated (***Figure 6—figure supplement 1***) and regression on both reporters again results in a correlation coefficient nearly equal to regressing on *Pmir-47*::GFP alone (***Figure 6c***, $R^2$ = 0.47). Semipartial correlation analysis was similar to the previous case (***Figure 6c***), indicating that *Pmir-240–786*::GFP is wholly redundant with *Pmir-47*::GFP but not vice-versa. We thus infer that, of the reporters we examined, *Pmir-47*::GFP expression either integrates information most broadly across lifespan-determining pathways, or reports on a step the farthest downstream in a single lifespan-determining pathway.

Unsurprisingly, *Pmir-240–786*::GFP and *Pmir-793*::GFP also were correlated and did not act as independent predictors of lifespan (***Figure 6d***; $R^2$ = 0.13 and 0.24 for *Pmir-240–786*::GFP and *Pmir-793*::GFP, respectively, with a joint $R^2$ = 0.25, which is close to the figure for *Pmir-793*::GFP alone and much less than 0.37, the sum of the two individual values). Confirmed by semipartial correlation analysis, this suggests that *Pmir-240–786*::GFP is almost wholly redundant with *Pmir-793*::GFP (save approximately 4% of lifespan variation which the former transgene can predict independent of *Pmir-793*::GFP). We infer that *Pmir-793*::GFP reports on information that is downstream of (or more broadly integrated than) *Pmir-240–786*::GFP, with respect to lifespan determination.

Overall, our data is consistent with a model wherein *Pmir-240–786*::GFP, *Pmir-793*::GFP, and *Pmir-47*::GFP expression provide hierarchical information about a lifespan-determining pathway or pathways. While there are multiple scenarios consistent with our results, all reflect the informational hierarchy depicted in ***Figure 6e*** (left), where *Pmir-47*::GFP provides strictly more information about lifespan than *Pmir-793*::GFP, which itself provides strictly more information than *Pmir-240–786*::GFP. Such a hierarchy is consistent with a scenario in which *Pmir-47*::GFP provides the most downstream report on a single lifespan-determination process (***Figure 6e***, center). It is also consistent with a scenario in which *Pmir-47*::GFP reports on three distinct lifespan-determination pathways, *Pmir-793*::

GFP reports on two of those, and *Pmir-240–786*::GFP on a single pathway (*Figure 6e*, right). Last, it could also be that each transgene reports on the exact same step of a single pathway, but that the different transgenes also reflect different degrees of extraneous, unrelated transcriptional input. Each model has in common, however, the key fact that the bulk of the three transgenes' correlation with lifespan is due to their mutual transcriptional relationship with a single lifespan-determining process.

It is particularly interesting in this context to recall that each of these reporters is expressed in distinct tissues; moreover, the expression trend of *Pmir-240–786*::GFP (increasing throughout life) is quite different from the other two reporters (which peak at mid-adulthood and decline). Nevertheless, the levels of these three reporters are all correlated (or anticorrelated) and provide largely redundant information about a single lifespan-determining process. Such redundancy and correlations among reporters expressed in distinct tissues could not exist without information flow between tissues, with the most likely candidate being a cell non-autonomous signaling process. We therefore conclude that *Pmir-47*::GFP, *Pmir-793*::GFP, and *Pmir-240–786*::GFP likely report on a global, cell non-autonomous (and DAF-16-independent) process of lifespan determination, which affects the entire animal's physiology.

## Discussion

In order to understand the relationship between inter-individual variation in transcriptional / gene-regulatory states and future lifespan, we used tools previously developed by our lab (*Zhang et al., 2016*; *Pittman et al., 2017*) to examine expression of 22 *PmiRNA*::GFP reporters (*Martinez et al., 2008*), across hundreds of individuals throughout life. This dataset provides a window into transgene-expression dynamics at a temporal and spatial resolution unprecedented for longitudinal experiments. Compared to previous work which followed a candidate-gene approach (*Pincus et al., 2011*; *Sánchez-Blanco and Kim, 2011*; *Rea et al., 2005*), this is the first systematic, unbiased examination of the relationship between promoter activity and future lifespan. Overall, we find that the ability to predict future lifespan is not a rare property, as 10 of the 22 transgenes tested were robustly and repeatedly correlated with individual longevity. We were not able to identify a clear pattern among lifespan-predictive vs. non-predictive miRNA promoters in terms of tissue of expression or temporal pattern in gene expression; instead, it appears that many transgenes, across many tissues, report on the activity of a smaller number of lifespan-determining processes. While previous studies have implicated inter-individual differences in IIS and more specifically, DAF-16 activity, in lifespan variability, we have identified at least one distinct, organism-wide transcriptional program that acts independently of *daf-16* in reflecting or determining future lifespan.

When measured while at least 90% of the population remained alive, 10 *PmiRNA*::GFP reporters met our criteria as predictors of longevity (*Figure 2*, *Table 2*), explaining between 17% and 33% of the variability in lifespan among genetically identical individuals. Overall, we found that longevity is generally associated with the preservation of young-adult reporter expression states. Among micro-RNA reporters that predict lifespan, those with high young-adult levels that decrease with age are positively correlated with longevity, such that long life is associated with high and/or non-decreasing GFP expression. Conversely, the single lifespan-predictive reporter that has low young-adult levels and increases with age is negatively correlated with lifespan; long life is associated with maintenance of low / non-increasing GFP levels. The fact that both an individual's average level of expression and the slope of expression over time can inform future lifespan suggests that both magnitude and maintenance of expression may be important for longevity. However, it is important to note that not all genes that change with time are predictive of future lifespan. For 12 of the 22 reporters examined, maintenance of 'youthful' expression patterns had no relationship with lifespan.

Nevertheless, we were surprised to find that nearly half of the *PmiRNA*::GFP reporters we tested *did* correlate with lifespan, especially as we did not take into account any prior association with aging or lifespan in selecting which reporters to examine. While lifespan-predictive reporters have previously been identified among hand-picked candidate genes (*Pincus et al., 2011*; *Sánchez-Blanco and Kim, 2011*), only unbiased screens such as the one we conducted can determine whether this is a relatively common or rare property. Overall, we conclude that it is not particularly uncommon or unusual for a transcriptional reporter to predict future lifespan, at least among those driven by microRNA promoters.

The ability to predict future lifespan is not a completely generic property of all GFP transgenes, however: in addition to the 12/22 the *PmiRNA*::GFP reporters that did not correlate with future lifespan, neither of the non-miRNA reporters we tested correlated strongly with lifespan (*Figure 2—figure supplement 1*); negative results for other transcriptional reporters have also been published (*Sánchez-Blanco and Kim, 2011*). These negative results also suggest that it is not simply the case that the ability to express *any* GFP transgene reflects a healthy, pro-longevity state. This conclusion is also bolstered by the observation that there exist promoter::GFP constructs whose expression *negatively* correlates with future lifespan (identified both in the present work and in previous publications *Pincus et al., 2011*). While recent work has shown that the capacity to fold and stabilize proteins may relate to the ability of *Phsp-16.2*::GFP to predict future lifespan after a heat shock (*Burnaevskiy et al., 2019*), our findings suggest that general protein expression capacity does not relate to future lifespan in un-heat-shocked animals.

More generally, similar logic suggests that none of the post-transcriptional steps that influence the measured GFP signal (e.g. the rates of mRNA degradation, translation, protein folding, protein degradation, and/or GFP photobleaching) explain the relationship between GFP levels and future lifespan among the strains we examined. This is because all of the *PmiRNA*::GFP reporters we examined produce identical mRNA transcripts; only the transcriptional control of these reporters differs. Thus, the salient difference between reporters that do vs. do not correlate with lifespan must necessarily be at the level of their transcriptional regulation.

Further study is required to identify the precise mechanism by which the transcription of these transgenes is regulated in a lifespan-predictive manner. One obvious possibility is transcription-factor binding in the promoter regions of the transgenes. However, position-specific effects are also plausible. Each multi-copy transgene is randomly integrated into the genome, and local regulatory sequences or chromatin accessibility can also influence the level of transgene expression. Regardless, our finding that many transgenes possess this property implies that is not an particularly rare property of promoter sequences, insertion site, or transgene structure.

Larger-scale studies will be necessary to determine whether a substantial fraction of promoters / integration sites across the genome act in a lifespan-responsive fashion, or whether this property is enriched at specific loci, or among promoters of regulatory genes like miRNAs or transcription factors.

No simple properties distinguish the *PmiRNA*::GFP reporters that predict future lifespan from those that do not. For every lifespan-predictive reporter, there are non-predictive reporters with nearly identical temporal trends in expression (*Figure 1*; *Table 2*), and non-predictive reporters with expression in the same tissues/organs (*Table 1*, *Figure 1—figure supplement 1*). This suggests that no single tissue is critical for individual lifespan, but instead that lifespan-determining processes act organism-wide, involving multiple tissues in a cell non-autonomous manner.

We next set out to determine the identity of the lifespan-determining cellular or genetic processes that these *PmiRNA*::GFP transgenes report on. The most straightforward hypothesis is that if expression of a *PmiRNA*::GFP reporter is positively correlated with lifespan, that is because the endogenous, reported-on miRNA acts to prolong lifespan. Several lines of evidence, however, suggest that this is not generally the case. First, the majority of predictive reporters we identified correspond to miRNAs with no published lifespan or aging phenotype (*Table 1*). *lin-4*, *mir-228*, and *mir-246* have been previously associated with longevity; knockout and overexpression studies have reported *lin-4* and *mir-246* as positive regulators of lifespan and *mir-228* as a negative regulator (*de Lencastre et al., 2010*; *Boehm, 2005*; *Smith-Vikos et al., 2014*). We found that as expected, reporters for *lin-4* and *mir-246* correlate positively with lifespan; surprisingly, *Pmir-228*::GFP also correlates positively with lifespan. It is important to note, however, that it should not be assumed that genes that change the population's mean lifespan when removed or overexpressed are necessarily those that act to determine inter-individual variation in lifespans among a genetically unperturbed population. In particular, the presence of regulatory feedback and buffering processes can dramatically complicate this relationship.

Regardless, a second line of evidence also suggests that the relationship between several of the *PmiRNA*::GFP reporters and lifespan is not via the activity of the endogenous, reported-on miRNA. Specifically, the correlation of two of the predictive reporters, *Pmir-243*::GFP and *Pmir-47::GFP*, with lifespan is not abrogated in the absence of the endogenous miRNA (*Figure 4*). Were these microRNAs a key part of the lifespan-determining transcriptional program reported on by the GFP levels,

the correlation between GFP levels and lifespan would have disappeared absent the microRNAs themselves. In addition, we also observed no substantial difference in lifespan between wild-type and *mir-243* and *mir-47* loss of function mutants.

These results indicate that neither *mir-243* nor *mir-47* play a direct functional role in determining lifespan in wild-type individuals. Our results are not surprising given most single miRNA (*Miska et al., 2007*) or even entire miRNA family (*Alvarez-Saavedra and Horvitz, 2010*) loss-of-function mutants are not required for normal development or viability; often phenotypes are only observed in sensitized backgrounds (*Brenner et al., 2010*) or when multiple members of a miRNA family are absent (*Brenner et al., 2012*). While *mir-243* is not reported to be part of a family, *mir-47* shares over 70% sequence identity with *mir-46* (*Ibáñez-Ventoso et al., 2008*); thus, redundancy between family members could mask a functional role for *mir-47* in aging.

Overall, this suggests that *Pmir-243*::GFP and *Pmir-47*::GFP (and potentially other of the lifespan-predictive reporters we identified) act as indirect, transcriptional reporters of some separate lifespan-determining process. Inspection of the promoter regions of these genes can provide hypotheses about the factors that may be both acting to determine lifespan and to control expression of these GFPs. While little is known in general about transcriptional control of miRNAs, much less the individual transcription factors involved (*Turner and Slack, 2009*), there is suggestive evidence. Yeast one-hybrid assays performed with *C. elegans* genes (*Fuxman Bass et al., 2016*) identify PQM-1, ELT-4, and HLH-30, transcription factors that are known to promote longevity (*Bansal et al., 2014*; *Tepper et al., 2013*; *Lin et al., 2018*), as likely to bind to the promoter sequence of *mir-243*. ZTF-8, a transcription factor that mediates the DNA damage response (*Kim and Colaiácovo, 2014*), may interact with the *mir-47* promoter.

Regardless of the immediate transcriptional control of *Pmir-243*::GFP and *Pmir-47*::GFP, it is nevertheless the case that the expression of these reporters must in some way be linked to one or more lifespan-determining processes or pathways. As the insulin/IGF-1-like signaling pathway (IIS), specifically the branch controlled by DAF-16, is arguably the best-studied 'longevity pathway' in *C. elegans*, we asked whether these transgenes predict lifespan by correlating with the level of DAF-16 activity. Because the reporters still predict lifespan in an IIS-deficient *daf-16* mutant, we conclude that this is not the case (*Figure 5*). This is surprising given that many other longevity-predictive reporters, such as *Pmir-71*::GFP and *Psod-3*::GFP, do require *daf-16* to correlate with future lifespan (*Pincus et al., 2011*; *Sánchez-Blanco and Kim, 2011*). (Note that these latter results are as expected, because the endogenous miRNA *mir-71*, interacts genetically with IIS to functionally influence lifespan [*de Lencastre et al., 2010*], and because *sod-3* is a direct transcriptional target of DAF-16 [*Sánchez-Blanco and Kim, 2011*]) Likewise the correlation between *Phsp-16.2*::GFP and lifespan after heat shock has also been proposed to be IIS-dependent (*Burnaevskiy et al., 2019*; *Mendenhall et al., 2017*). However, despite the importance of DAF-16 in determination of lifespan, both at a population and an individual level, *Pmir-243*::GFP and *Pmir-47*::GFP report on one or more processes or pathways that are genetically distinct from *daf-16*.

Beyond the precise identity of the lifespan-determining processes that are reported on by these *PmiRNA*::GFPs, we also attempted to estimate the overall number of such processes that might be at play. Because the correlation coefficients sum to greater than one (*Table 2*), some of the *PmiRNA*::GFPs must redundantly report on the same processes. We thus wondered whether we had identified 10 transgenes that each served as noisy measures of one single process (one extreme), or reporters for six to eight independent lifespan-determining processes. Fortunately, several of the highly lifespan-predictive *PmiRNA*::GFPs (including *Pmir-47*::GFP, which we had extensively analyzed as above) were expressed in distinct, non-overlapping tissues. This allowed us to simultaneously measure the lifespan-predictive signal from pairs of reporters in the same individuals. We found that *Pmir-793*::GFP and *Pmir-240–786*::GFP did not provide any information about future lifespan that was not already captured by *Pmir-47*::GFP expression. Furthermore, *Pmir-240–786*::GFP was completely redundant with *Pmir-793*::GFP. As such, all three GFPs must necessarily report at least in part on a single lifespan-determining process. This redundancy stands in stark contrast with the fact that these genes are expressed in very distinct tissues (*Table 1*; *Figure 1—figure supplement 1*) and with distinct temporal expression patterns (*Figures 1* and *2*). This suggests that *Pmir-47*::GFP, *Pmir-793*::GFP, and *Pmir-240–786*::GFP report on a cell non-autonomous process that affects and integrates among multiple tissues throughout aging.

High-resolution analysis of lifespan curves has suggested that a single state of 'biological resilience' emerges from the interactions of multiple, densely-interlinked lifespan-determining genes/processes (*Stroustrup, 2016*). Such a state of resilience might be expected to be cell- and tissue-autonomous; if so, it is possible that these *PmiRNA*::GFPs report on some aspects of overall resilience or biological reserve. Note, however, that *Pmir-47*::GFP expression, measured specifically from the vulva, does provide significant prediction of future lifespan that is not captured by *Pmir-793*::GFP or *Pmir-240–786*::GFP. Thus, in addition to shared cell and/or tissue non-autonomous processes, *Pmir-47*::GFP may also report on separate, tissue-specific processes. Indeed, vulval tissue integrity has been identified as an important determinant of health and survival *C. elegans* (*Leiser et al., 2016*).

We set out to explore the hypothesis that inter-individual differences in lifespan arise from fate-commitment-like mechanism, whereby certain individuals persistently maintain transcriptional states that assure future stress resistance and extended lifespan. Overall, our results clearly demonstrate that individuals with distinct future fates can in fact be distinguished on the basis of gene-regulatory states (as read out by a number of *PmiRNA*::GFP reporters). A developmental, or even young-adult, origin to these transcriptional states we identified appears unlikely, however: none of the reporters were particularly predictive of lifespan before approximately the third day of adulthood. Despite substantial inter-individual variation in GFP expression early in adulthood (*Figure 1—figure supplements 2* and *3*), the window in which that expression variability correlates with remaining lifespan begins roughly a day or so before the start of the population die-off and ends when approximately 25% of the population remains alive (*Figure 3*). This suggests that the lifespan-determining transcriptional processes that the *PmiRNA*::GFPs report on are set relatively late in life. This contrasts with the early-adult window in which major changes to the stress-response machinery take place (*Labbadia and Morimoto, 2015*), which we had originally suspected might be a key inflection point at which inter-individual differences in lifespan-determining stress-resistance pathways emerge.

The hypothesis that gene-regulatory programs determine individual lifespan should be compared against the null hypothesis that inter-individual variability in lifespan is solely due to stochastic differences in accumulation of damage throughout life. It is possible that the lifespan-predictive *PmiRNA*::GFPs we identified are simply very early reporters of the transcriptional consequences of lifespan-limiting stochastic damage (such as pathogenesis [*Zhao, 2017*; *Podshivalova et al., 2017*] or intestinal permeabilization [*Gelino et al., 2016*]). This would be consistent with 'youthful' gene-expression levels predicting long life: a change in gene expression would be indicative of an individual having received a potentially lethal insult. Alternately, and equally consistent with our data, it may be that loss of youthful, protective gene-expression states is a risk factor for suffering exogenous insults (e.g. for becoming bacterially infected), or for being unable to recover from such insults (e.g. for unsuccessfully countering a bacterial infection). In this scenario, differences in the timing of the departure from an optimal gene-expression program drives differences in individual lifespan. The fact that the reporters remain somewhat predictive of future lifespan even when excluding individuals that die within the subsequent 24–48 hr (*Figure 3—figure supplement 1*) provides some evidence for this latter view. However, it is also possible that the transgenes report on stochastic damage that leads only slowly to death. Further genetic and functional analysis of these lifespan-predictive reporters will be necessary to conclusively determine whether they lie upstream or downstream of exogenous, lifespan-limiting insults.

Overall, we find that it is quite common for microRNA-promoter::GFP transgene expression to predict an individual's future lifespan, at mid-to-late adulthood. Moreover, at least three and potentially as many as all 10 of the 22 transgenes we examined ($\approx 15$–50%) redundantly report on a single, cell non-autonomous, IIS-independent, organismal state that relates to future lifespan. This state (whatever its specific molecular identity) is thus likely to be highly connected to many aspects of organismal biology.

Ultimately, this work raises a host of questions. What is the origin and identity of this state? Is it early responses to fatal damage? A state that controls the risk of suffering such damage? What specific transcription factors / chromatin accessibility changes define this state, and/or act on these transgenes in a lifespan-predictive fashion? We believe that answering these questions will shed light on the overall nature of the aging process, and identify gene regulators that not only identify prospectively short-lived individuals, but can be manipulated to switch those individuals toward long-lived fates.

## Materials and methods

### Strains

The following strains were obtained from the Caenorhabditis Genetics Center (CGC): VT1735 (*Pmir-788*::GFP), VT1541 (*Pmir-360*::GFP), VT1733 (*Pmir-60*::GFP), VL405 (*Pmir-63*::GFP), VL440 (*Pmir-47*::GFP), VT1153 (*Plet-7*::GFP), VT1072 (*Plin-4*::GFP), VL412 (*Pmir-79*::GFP), VT1379 (*Pmir-59*::GFP), VT1474 (*Pmir-243*::GFP), VT1189 (*Pmir-241*::GFP), VT1607 (*Pmir-246*::GFP), VT1485 (*Pmir-228*::GFP), VT1600 (*Pmir-85*::GFP), VT1481 (*Pmir-51*::GFP), VT2020 (*Pmir-793*::GFP), VT2021 (*Pmir-794*::GFP), VL370 (*Pmir-240–786*::GFP), VT1470 (*Pmir-242*::GFP), VT1160 (*Pmir-84*::GFP), VT1589 (*Pmir-90*::GFP), VT1665 (*Pmir-1*::GFP), PD4793 (*Pmyo-2*::GFP; *Ppes-10*::GFP; *F22B7.9*::GFP), B010652 (*Pcpna-2*::GFP) CF1038 (*daf-16(mu86)*), MT15454 (*mir-243(n4759)*), VC328 (*mir-47(gk167)*).

All miRNA reporter (*PmiRNA*::GFP) strains were selected from a larger library originally created by *Martinez et al., 2008*. Briefly, miRNA promoters were defined as 300 to 2000 base pair regions upstream of the mature miRNA or stem-loop sequence and cloned via Gateway cloning. The GFP and 3′ UTR for the constructs were cloned from the 'Fire Lab Vector Kit' construct pPD95.75, containing a GFP with artificial introns and an *unc-54* 3′ UTR. The transgenic strains were created via microparticle bombardment to generate low-copy random integrants, with UNC-119 rescue used as a selection marker.

All reporter strains were crossed into BA671 (*spe-9(hc88)*), a temperature-sensitive sterile mutant with a wild-type lifespan at 25.5 °C (*Fabian and Johnson, 1994*). Strains were maintained at 20°C and for all assays embryos were shifted to 25°C to prevent reproduction in the single-animal culture apparatus.

### Single-animal culture

For longitudinal analysis, *C. elegans* were reared in high-density single-animal culture devices ('worm corrals'), which we have described previously (*Zhang et al., 2016*; *Pittman et al., 2017*). In brief, 8-armed PEG-thiol (Jenkem Technology) and PEG-diacrylate (Sigma-Aldrich) are dissolved in a modified nematode growth medium (NGM) at concentrations of 133 mg/mL and 37 mg/mL, respectively, and filter-sterilized. The PEGs are mixed in a 1:1 ratio and cholesterol in ethanol is added to a final concentration of 4 µL/mL. The final media is poured into an aluminum frame adhered to a glass slide with PDMS (Dow Corning) and allowed to cure into a solid hydrogel at room temperature for approximately 2 hr. *E. coli* OP50-1, resuspended to 50% w/v from an overnight culture, is deposited into an array of 0.4 µL droplets onto the gel. A pretzel-stage embryo is placed into each droplet via eyelash pick. Approximately 1 mL liquid PDMS is poured onto the gel; the PDMS crosslinks to the gel and cures into a solid form within 48 hr. The culture devices were housed on a microscope stage (Leica) in a custom-built climate chamber held at 25°C and 90% relative humidity.

### Image acquisition

Images of individual animals were automatically acquired every 4 hr at ×5 magnification using custom-built image acquisition and autofocus software. A bright-field, autofluorescence, and fluorescence image were taken of each animal at each timepoint. Fluorescence images were taken with a DAPI/FITC/TRITC filter (Semrock, DA/FI/TX-3X-A-000) and Lumencor Spectra X light source and were used to measure accumulation of autofluorescence material and transgene expression. For autofluorescence, an exposure time of 70 ms and 556/20 nm (center wavelength/bandwidth) excitation filter was used; fluorescence in these red emission wavelengths increases linearly with age and has been shown to correlate well with remaining lifespan (*Pincus et al., 2016*). An exposure time of 30–70 ms was manually chosen for each reporter strain (as strains vary considerably in brightness), with 40 ms most commonly used. Exposure times were consistent for all images within a given strain. A 480/17 nm excitation filter was used for all GFP imaging. Flat-field images of a fluorescent slide (Chroma) were collected before every image acquisition to control for spatial variation in illumination.

### Image measurement

All images were corrected for sensor noise (dark-current) and spatial variation in illumination (flat-field). Images were automatically segmented using a convolutional neural network to determine

pixels contained within the worm (*Ghiasi and Fowlkes, 2016*). Whole-animal summary statistics (e.g. 99th percentile intensity) were calculated from the fluorescence intensity distribution of these pixels. All measurements were performed in this fashion unless specifically indicated. Summary statistics are often generated by simply taking the average fluorescence intensity within the worm region. However, because the reporters we examined vary greatly in spatial and temporal expression pattern as well as absolute intensity, we found that no single summary statistic, such as the mean, was appropriate in all cases to robustly capture expression levels of each different reporter, or to distinguish GFP from background autofluorescence. For example, while taking the mean intensity across all 'worm pixels' can often reliably measure the fluorescence of a bright *PmiRNA*::GFP reporter expressed in a large tissue such as the intestine, the same mean may mostly capture intestinal autofluorescence for a reporter that is dim, or expressed only in a smaller tissue like the pharynx or vulva (*Figure 1—figure supplement 1*). Furthermore, expression of most reporters decreases with time while autofluorescence increases, further challenging robust, automated detection of reporter expression with measurements like mean intensity. Thus, we calculated a number of potential summary statistics, including 95th percentile intensity, the mean of pixels over the 99th percentile intensity, and maximum intensity (see Materials and methods), and for each reporter selected the one that most reliably captured *bona-fide PmiRNA*::GFP expression in both young and aged animals (*Table 2*). Our selection was based on manual comparisons of images of reporter expression with maps identifying the set of pixels measured by each summary statistic (i.e. if the pixels at the 95th-percentile intensity level were never co-localized with reporter-gene expression, the 95th percentile intensity was not considered an appropriate summary statistic). *Figure 1—figure supplement 1* shows examples of such maps. Among plausible summary statistics, we selected those which produced the least measurement noise timepoint-to-timepoint.

For some strains with dim, spatially restricted fluorescence, measurements were performed on specific structures (e.g. head) rather than whole-worm images to avoid inadvertently measuring autofluorescence. We manually annotated the center of these structures from brightfield images and extracted summary measurements from fluorescence-image pixels within a circle of defined radius from those center-points. For *Pmir-1*::GFP, a circle with a radius of 5 pixels (6.5 µm) centered at the pharynx was used. For *Pmir-241*::GFP, a circle with a radius of 25 pixels (32.5 µm) centered at the pharynx was used. For *Pmir-59*::GFP, a circle with a radius of 25 pixels (32.5 µm) centered at the vulva was used to extract expression measurements.

For dual reporter strains, *Pmir-47*::GFP and *Pmir-793*::GFP expression was extracted from manually annotated circular regions centered at the vulva and pharynx. These are the predominant areas of detectable expression, especially in aged animals. *Pmir-240–786*::GFP expression was measured from areas of the worm excluding the manually annotated region for *Pmir-47*::GFP or *Pmir-793*::GFP. Varying the radius of the circular region did not substantially affect correlation with lifespan (data not shown).

Time of hatch, first egg lay, and death were annotated manually for all animals to determine lifespan. Individuals that hatched prior to the start of the experiment were annotated as hatching at t = 0; this timepoint is within at most 3 hr the actual hatch time (*Zhang et al., 2016*). No more than 5% of eggs in a given experiment fail to hatch in our system; often all hatch successfully. Of those that do hatch, <5% (typically 1–2%) are excluded from further imaging and analysis due to production of offspring (the majority of the excluded individuals) or larval arrest (a small minority). In a minority of cases, additional individuals were excluded due to localized failure of the 'worm corral' device: mold contamination and/or gel desiccation. The remaining individuals were indistinguishable from those in other biological replicates.

All fluorescence measures made for every animal used in all figures and tables (and associated life-stage annotations) are provided in easily parsed text files as *Source data 1*.

## Data and statistical analysis

For visualizing population-level or cohort-level expression over time, each biological replicate was rescaled by median expression intensity and median lifespan before pooling. Expression data were fit to trendlines using LOWESS smoothing to facilitate visualization of expression levels over time. For regression and other correlative analyses, all expression values were z-transformed relative to each biological replicate and time rescaled by median lifespan of the replicate to facilitate data

pooling and prevent replicate-dependent effects. Regression results from raw data from each biological replicate are consistent with the pooled analyses (table S6 in *Supplementary file 1*).

Single and multivariable regression of biomarkers versus lifespan was performed using ordinary least-squares regression, and the coefficient of determination ($R^2$) calculated according to the standard formula. We aimed for a minimum sample size of 80 individuals, which leads to a statistical power of 95% to identify a true correlation of $R^2 > 0.15$ with $p < 0.05$. We have found empirically that below $R^2 = 0.15$, correlations with future lifespan are not generally robust across experimental replicates. Multivariable regressions for *PmiRNA*::GFP reporters were performed using both average expression and the slope of a least-squares linear fit line to an individual's expression data points in a defined time window. Confidence intervals for correlation coefficients were determined by bootstrap analysis. Each pooled reporter sample was resampled with replacement 1000 times and regressions were performed on the resampled data. The 2.5th and 97.5th percentiles of the resampled correlation coefficients were used to construct the 95% confidence interval.

Individuals were excluded from analysis if they did not survive through the entire time window under consideration. Unless otherwise noted, time windows were chosen such that 10% or less of the total population was excluded.

## Acknowledgements

We thank Tim Schedl, Drew Sinha, Will Pittman and Eric Kim for helpful discussions and thoughtful feedback on the manuscript. This work was supported by NIH grants T32 HG000045 and R01 AG057748, and a Beckman Young Investigator award from the Arnold and Mabel Beckman Foundation. Some strains were provided by the CGC, which is funded by the NIH Office of Research Infrastructure Programs (P40 OD010440).

## Additional information

### Funding

| Funder | Grant reference number | Author |
|---|---|---|
| National Institute on Aging | R01 AG057748 | Matthew C Mosley<br>Zachary Pincus<br>Isaac B Plutzer |
| National Human Genome Research Institute | T32 HG000045 | Holly E Kinser |
| Arnold and Mabel Beckman Foundation | Beckman Young Investigator | Holly E Kinser<br>Zachary Pincus |

The funders had no role in study design, data collection and interpretation, or the decision to submit the work for publication.

### Author contributions

Holly E Kinser, Conceptualization, Formal analysis, Validation, Investigation, Visualization, Methodology, Writing - original draft, Writing - review and editing, Designed the experiments, conducted the experiments, analyzed the data and wrote the data analysis software; Matthew C Mosley, Investigation, Writing - review and editing, Conducted the experiments; Isaac B Plutzer, Data curation, Investigation, Conducted the experiments; Zachary Pincus, Conceptualization, Software, Supervision, Funding acquisition, Methodology, Writing - original draft, Project administration, Writing - review and editing, Designed the experiments, wrote the data analysis software, wrote the image acquisition and image analysis software

### Author ORCIDs

Isaac B Plutzer http://orcid.org/0000-0002-9370-7763
Zachary Pincus https://orcid.org/0000-0001-9785-5977

Decision letter and Author response

Decision letter https://doi.org/10.7554/eLife.65026.sa1

Author response https://doi.org/10.7554/eLife.65026.sa2

# Additional files

## Supplementary files

- Source data 1. Source data for all figures.

- Supplementary file 1. Supplementary tables.

- Transparent reporting form

## Data availability

All data generated or analysed during this study are included in the manuscript and supporting files. Source data files have been provided for all figures and tables.

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
