## [Decision Letter]

[Editors’ note: the authors submitted for reconsideration following the decision after peer review. What follows is the decision letter after the first round of review.]

Thank you for submitting your work entitled "Global, cell non-autonomous gene regulation drives individual lifespan among isogenic *C. elegans*" for consideration by *eLife*. Your article has been reviewed by a Senior Editor, a Reviewing Editor, and three reviewers. The reviewers have opted to remain anonymous.

Our decision has been reached after consultation between the reviewers. Based on these discussions and the individual reviews below, we regret to inform you that your work will not be considered further for publication in *eLife*.

The reviewers agreed that the question posed is an important one and that some of the data presented is interesting. However, in the reviews and during the subsequent discussions, there was also a general consensus that several of the major claims about the causal structure of the ageing process and about the cell non-autonomous nature of the effects are not sufficiently supported in the current version of the manuscript. The reviewers suggested several additional experiments and controls that might be employed to address these limitations. However, given the COVID-19 situation, we appreciate that it may not be possible to carry out these additional experiments in a timely manner. Unfortunately, without substantial additional evidence, these claims would have to be removed or substantially weakened, necessitating changes that would go beyond major revisions.

The reviewers suggested that, should you decide to carry out the additional experiments and address these concerns in the future, you may consider re-submitting the manuscript as a new submission to *eLife*.

Reviewer #1:

This manuscript presents a very interesting study of the stochastic nature of aging. It has long been known that identical clones have varying lifespans as can be seen from every "control" lifespan curve published. The authors use a very innovative approach of tracking the expression of GFP marking miRNAs in individual worms to see if they are predictive of lifespan. The findings are equally striking as ~half of miRNAs correlate with lifespan.

The major issue with this manuscript is the way that it is written. The main result is striking, but the writing takes the reader into detailed descriptions of the statistical methods, counterfactuals and nearly philosophical discussions. The Results section needs to be cut. The Discussion needs to be cut. The authors should consider moving some of the text to the Materials and methods section.

I do not believe further experiments are needed.

Reviewer #2:

Kisner et al., present a thoughtful and carefully executed study of a panel of in vivo GFP-promoter fusion constructs that they deploy to dissect the causal structure of lifespan-determining processes in *C. elegans*. The authors focus on inter-individual differences within isogenic populations. They consider the covariation structure among their GFP reporters and lifespan and cleverly argue that inter-individual differences in lifespan are driven largely (but not completely) by some upstream factor shared among the various gene regulatory elements governing microRNA expression. Because these gene regulatory elements act in separate tissues and at different times in life, the authors conclude that there is, minimally, a hierarchical structure among lifespan-determining processes and, perhaps, that the lifespan of their animals was determined by a single organism-wide, cell-non-autonomous process. This seems like an important contribution as it articulates a precise question: what is this lifespan-determining process? The authors exclude one obvious answer-DAF-16 activity -and leave a compelling puzzle ripe for future study.

By employing bespoke quantitative approaches for GFP reporter and extensive manual validation, the authors decouple image processing quirks from their downstream statistical analyses. This is satisfying and likely contributes to the agreement between their measurements and the limited previously published results on the subject.

Much of the reasoning in this paper depends on comparisons between small correlations and r-squared values. In such contexts where "effect sizes" are small relative to the "noise", statistical analyses must be performed and interpreted with great care. The authors do this well, but they may have overlooked the potential for truncation effects to be driving some of the observed correlation between reporter measurements and lifespan. If present, truncation effects could potentially confound the major conclusions of the manuscript. The authors should address this concern (described in Major Point #1) with additional statistical analysis that rule out the possibility for truncation effects influencing their conclusions, or, if possible, by performing orthogonal experiments that support the same conclusions in a different way.

1) For each microRNA, the authors identify a time interval (shared among all individuals) during which they calculate the mean and change in pixel intensity data. To identify predictors of lifespan, they then regress these means and slopes against individual lifespan. If no deaths occurred during the time interval chosen, this would be a relatively straightforward calculation, as there would be no relationship between death time and the number of measurements used to calculate pixel intensity averages. However, the authors use time intervals that do include deaths, which means that short-lived animals will necessarily have fewer measurements available during the interval compared to longer-lived animals. This introduces a potential confounding effect that in principle could make time-dependent processes completely unrelated to lifespan appear like biomarkers.

Perhaps with similar concerns in mind, the authors state that they restrict the time intervals used for each reporter to a maximum upper bound of 90% survival. In Figure 2A, 90% survival appears to correspond to 8 days post-hatch. An upper bound at 90% of survival would reduce the effect of truncations but not eliminate them. Yet, the authors do not appear to follow their stated limitation, as in Table 2 the time windows appear to extend out to 10 days at which point 50% of individuals have died. This opens the possibility that truncation effects could underlie a substantial fraction of the reporter / lifespan correlations reported. Worse, because the death of an individual will necessarily truncate all reporters simultaneously, truncation effects could additionally contribute to the overlap in the predictive powers observed between multiple reporters.

2) The authors present two opposing models regarding the source of inter-individual differences: exogenous factors acting differentially between individuals throughout life and exogenous factors that establish stable inter-individual differences early in life. The contrast between stable changes early and continuous changes seems reasonable-but why must inter-individual different arise solely from exogenous factors? It would seem that the authors conceptual and analytic framework would describe equally well extrinsic and intrinsic factors (including spontaneous differences arising during development, or random stochastic fluctuations in gene regulatory networks occurring throughout life).

Reviewer #3:

In this paper, Holly Kinser, Matthew Mosley, Isaac Plutzer, and Zachary Pincus set out to identify biomarkers of aging in *C. elegans*. The questions they set out to answer are important to understanding the causal structure of the aging process. *C. elegans* is a great system to address these questions in a multicellular setting. The authors use a powerful imaging platform that the lab developed that enables them to image fluorescent reporters in individual worms throughout their lifespan. They focus on a set of micro RNA expression reporter fusions and identify several reporters with age-dependent changes in expression that correlate with remaining lifespan. For a subset of these reporters that they identify as strong predictors of life expectancy they find that these predictions remain in mutants of daf-16/FOXO, an important effector of the insulin singling pathway. The authors then examine whether multiple markers provide additional information about life expectancy and find that they do not. Overall, my impression is that these findings are interesting, but more work would be needed to ascertain the robustness and generality of these reporter predictions. In addition, some the central claims of the paper need better support.

1) How robust are these findings? The authors only use one transgenic line for each reporter. Additional lines of evidence are needed to determine if these findings are robust. This is important because, in principle, their findings could be dependent on the position in the genome where the transgene was integrated. That is, are the regulatory sequences in the transgenes sufficient to produce the age dependent expression patterns and correlations with life expectancy? Or could regulatory regions near the site of integration be necessary or sufficient for that regulation?

2) The authors take advantage of many existing reporter constructs fusing regulatory regions of micro RNA genes to GFP. They do not provide sufficient information about how these key tools were built. What are the promoter sequences used? Did the GFP reporter have introns? Which 5' UTR was used in each case? What coinjection marker was used in each case? Where the transgenes single copy or multicopy? Where the lines outcrossed? This is important because it is possible that the age-dependent regulation they observe could be due to factors other than the promoter sequences. For example, the unc-54 3' UTR is well known to drive gene expression in the posterior intestine, and introns in GFP stabilize the mRNA.

3) The authors make several claims that need better support. They claim that age-dependent predictors are "independent of insulin signaling". But they only look at the dependency for one effector of this pathway, DAF-16, when there is extensive literature going back decades showing that SKN-1/NRF, HSF-1/HSF, and many other transcription factors also act downstream of the DAF-2 receptor (and independently of DAF-16) in the regulation of lifespan. Could those effectors of insulin signaling be involved?

4) The authors claim in the title that "Global, cell non-autonomous gene regulation drives individual lifespan." Yet this set of claims is one of many models that can explain their findings. Issues with these specific claims:

4.1) Gene regulation: What is the evidence that transcriptional changes cause differences in lifespan? The authors say in the results that all their models share a "mutual transcriptional relationship with a single lifespan-determining process." Does that process need to be transcriptional just because it also affects expression of the GFP reporters?

4.2) Cell non-autonomous: I am not sure what the rationale for this claim is. Why couldn't there be a set of cell autonomous killing processes in the different tissues where the separate predictors of lifespan are expressed? This would be similar to links in a chain, the chain breaks when the weakest link breaks, but links break independently (autonomously). What would the authors have expected to turn out differently if the lifespan determining process acted cell autonomously?

5) The authors claim in the Abstract that they "demonstrate a hierarchy among several transgenes expressed in distinct tissues" that report on a single lifespan determining process. Could the transgenes with non-additive lifespan predicting capacity just have overlapping independent (ie non-hierarchical) transcriptional determinants? (that is could they share regulatory elements that each report on an independent lifespan determinant?)

---

## [Author Response]

[Editors’ note: the authors resubmitted a revised version of the paper for consideration. What follows is the authors’ response to the first round of review.]

Reviewer #1:This manuscript presents a very interesting study of the stochastic nature of aging. It has long been known that identical clones have varying lifespans as can be seen from every "control" lifespan curve published. The authors use a very innovative approach of tracking the expression of GFP marking miRNAs in individual worms to see if they are predictive of lifespan. The findings are equally striking as ~half of miRNAs correlate with lifespan.The major issue with this manuscript is the way that it is written. The main result is striking, but the writing takes the reader into detailed descriptions of the statistical methods, counterfactuals and nearly philosophical discussions. The Results section needs to be cut. The discussion needs to be cut. The authors should consider moving some of the text to the Materials and methods section.

We have read over the manuscript with a critical eye and shortened explanatory text and/or moved text to the Materials and methods section where applicable. We did our best to maintain a balance between brevity and providing sufficient detail about analytic methods that are not common in the field. In particular, we have retained some of the counterfactuals in the description of our statistical pathway analyses. We feel that these are the best way to clearly address questions of “how would the results have differed had an alternate hypothesis been true instead?”, such as raised by reviewer #3. Overall, we believe the revised manuscript is much clearer.

Reviewer #2:[…]Much of the reasoning in this paper depends on comparisons between small correlations and r-squared values. In such contexts where "effect sizes" are small relative to the "noise", statistical analyses must be performed and interpreted with great care. The authors do this well, but they may have overlooked the potential for truncation effects to be driving some of the observed correlation between reporter measurements and lifespan. If present, truncation effects could potentially confound the major conclusions of the manuscript. The authors should address this concern (described in Major Point #1) with additional statistical analysis that rule out the possibility for truncation effects influencing their conclusions, or, if possible, by performing orthogonal experiments that support the same conclusions in a different way.1) For each microRNA, the authors identify a time interval (shared among all individuals) during which they calculate the mean and change in pixel intensity data. To identify predictors of lifespan, they then regress these means and slopes against individual lifespan. If no deaths occurred during the time interval chosen, this would be a relatively straightforward calculation, as there would be no relationship between death time and the number of measurements used to calculate pixel intensity averages. However, the authors use time intervals that do include deaths, which means that short-lived animals will necessarily have fewer measurements available during the interval compared to longer-lived animals. This introduces a potential confounding effect that in principle could make time-dependent processes completely unrelated to lifespan appear like biomarkers.

Reviewer #2 is correct that death during the analysis window can introduce confounding effects, and this is something we tried to consider carefully. For the analysis referred to here (Figure 2 and Table 2), animals that died within the time-window had been excluded in order to avoid the truncation effects described. We have modified the text of the methods section to emphasize this point, as the previous language was unclear.

In addition, Figure 3—figure supplement 1 shows the same results when censoring individuals that go on to die within one or two days *after* the time-window, to entirely remove potential end-of-life effects. This does not materially alter the conclusions presented. We have amended the main text related to Figure 3—figure supplement 1 in order to make this point more clear.

Nevertheless, we acknowledge that such censorship itself can introduce subtle biases. In Figure 2 and Table 2, the analysis does not include any time windows in which more than 10% of the population dies (and is thus censored). To determine whether this 10% threshold itself drives any of our results, we have now included a new Supplementary file 2, repeating this analysis with a 5% threshold. We found that the predictive power of the tested *PmiRNA*::GFP reporters was largely unchanged by the reduction in censorship. (Further reducing the threshold is impractical, as even relatively rare early mortality events prevent such time windows from extending much past early adulthood.)

Perhaps with similar concerns in mind, the authors state that they restrict the time intervals used for each reporter to a maximum upper bound of 90% survival. In Figure 2A, 90% survival appears to correspond to 8 days post-hatch. An upper bound at 90% of survival would reduce the effect of truncations but not eliminate them. Yet, the authors do not appear to follow their stated limitation, as in Table 2 the time windows appear to extend out to 10 days at which point 50% of individuals have died. This opens the possibility that truncation effects could underlie a substantial fraction of the reporter / lifespan correlations reported. Worse, because the death of an individual will necessarily truncate all reporters simultaneously, truncation effects could additionally contribute to the overlap in the predictive powers observed between multiple reporters.

We observe differences in lifespans across our experiments, primarily due to per-strain effects, and to a lesser extent to the batch/seasonal effects that all *C. elegans* lifespan labs confront. As shown by the small arrows in Figure 3, the 90% survival times differ somewhat across strains. The time windows in Table 2 are thus based on the empirical 90% survival time for each strain in question. We have clarified these points in the text.

2) The authors present two opposing models regarding the source of inter-individual differences: exogenous factors acting differentially between individuals throughout life and exogenous factors that establish stable inter-individual differences early in life. The contrast between stable changes early and continuous changes seems reasonable-but why must inter-individual different arise solely from exogenous factors? It would seem that the authors conceptual and analytic framework would describe equally well extrinsic and intrinsic factors (including spontaneous differences arising during development, or random stochastic fluctuations in gene regulatory networks occurring throughout life).

We completely agree, and did not intend to rule out intrinsic factors. The working model that we think most plausible is that transient fluctuations in regulatory activity can, in certain cases, lead to stable differences in gene expression – perhaps via the establishment of positive feedback loops. These initial transient events may well arise from either intrinsic *or* extrinsic causes, or some mix; we don’t currently have a means to distinguish these cases. We have attempted to make the text more clear on this matter.

Reviewer #3:In this paper, Holly Kinser, Matthew Mosley, Isaac Plutzer, and Zachary Pincus set out to identify biomarkers of aging in *C. elegans*. The questions they set out to answer are important to understanding the causal structure of the aging process. *C. elegans* is a great system to address these questions in a multicellular setting. The authors use a powerful imaging platform that the lab developed that enables them to image fluorescent reporters in individual worms throughout their lifespan. They focus on a set of micro RNA expression reporter fusions and identify several reporters with age-dependent changes in expression that correlate with remaining lifespan. For a subset of these reporters that they identify as strong predictors of life expectancy they find that these predictions remain in mutants of daf-16/FOXO, an important effector of the insulin singling pathway. The authors then examine whether multiple markers provide additional information about life expectancy and find that they do not. Overall, my impression is that these findings are interesting, but more work would be needed to ascertain the robustness and generality of these reporter predictions. In addition, some the central claims of the paper need better support.1) How robust are these findings? The authors only use one transgenic line for each reporter. Additional lines of evidence are needed to determine if these findings are robust. This is important because, in principle, their findings could be dependent on the position in the genome where the transgene was integrated. That is, are the regulatory sequences in the transgenes sufficient to produce the age dependent expression patterns and correlations with life expectancy? Or could regulatory regions near the site of integration be necessary or sufficient for that regulation?

We believe that our central claims are robustly supported. However, these claims are somewhat different in nature from much other work in the area, even previous work on fluorescent predictors of future lifespan, so it’s worth unpacking things a bit.

For all the reasons that the reviewer cites, we agree that our results *do not* robustly support claims such as, for example, “transcription factor binding activity in the promoter region 2 kb upstream of *mir-239* is sufficient to predict future lifespan”. If one were to view the manuscript as essentially a collection of 10 independent claims along these lines, for 10 different microRNA promoter regions, then the reviewer’s concerns would be well warranted.

We feel, however, that the more interesting – and better supported – conclusions from our work derive from the aggregate analysis of all the transgenes. In particular, the central and novel claim we advance is that long- and short-lived individuals are pervasively different from one another in terms of gene regulation and expression. In support of this, we marshal evidence that there exist one or more gene-regulatory processes in *C. elegans* that (a) reflect future lifespan, and (b) act on a wide variety of genes (in this case, half of the library of transgenes tested). Indeed, one of the lifespan-predictive regulatory process we found appears to act on at least three separate transgenes.

It is entirely plausible that these regulatory processes act on the transgenes, at least in part, via position-specific effects (e.g. local chromatin accessibility, or endogenous regulatory sequences at the integration sites). However, this possibility does not conflict with the central claim above. We would still find it surprising and interesting to learn that, for example, half of all possible integration sites (and thus, perhaps half of all genomic loci) confer lifespan-predictive properties on genes expressed therefrom.

We are of course acutely interested in the molecular identity of these yet-unknown regulatory processes; as such we are actively attempting to identify specific regulatory sequences or positional effects that confer lifespan-predictive abilities on these transgenes. But these questions are distinct from the claims made in this manuscript.

Again, we absolutely agree that our manuscript should not be interpreted as making strong claims about the lifespan-predictive power of particular promoter sequences per se. In the revised text we have redoubled our efforts to stay away from claims about promoter sequences, and instead focused solely on properties of the specific transgenes we examined. If anything, we feel that the central finding (that fully half of a library of transgenes, not pre-selected to be relevant to aging, can nevertheless predict lifespan) is more interesting than just adding to the number of known lifespan-predictive reporters in *C. elegans* (i.e. extending the work in Pincus et al., 2011 and Sanchez-Blanco et al., 2011).

Why do we believe our central claim is robustly supported? First, the fact that multiple independently integrated transgenes predict future lifespan indicates that this is not a rare property, nor one that is exquisitely dependent on a specific insertion site or a specific interaction between insertion site and transgene. Second, we identified three separate transgenes that all report on the same lifespan-predictive regulatory process. That is, the existence and activity of this specific process (whatever that process actually is) is attested to by multiple independently constructed transgenic lines. This also shows that the lifespan-predictive property is not entirely idiosyncratic and specific to each transgene, and further demonstrates the degree to which lifespan-predictive processes regulate genes across tissues, promoter sequences, and integration sites.

Last, the reviewer asks, “are the regulatory sequences in the transgenes sufficient to produce the age dependent expression patterns?” This does appear to be the case; as noted in the manuscript, the expression patterns we identify are completely consonant with those in Kato et al., 2009, which more coarsely characterized miRNA expression dynamics over aging using small-RNA-seq.

2) The authors take advantage of many existing reporter constructs fusing regulatory regions of micro RNA genes to GFP. They do not provide sufficient information about how these key tools were built. What are the promoter sequences used? Did the GFP reporter have introns? Which 5' UTR was used in each case? What coinjection marker was used in each case? Where the transgenes single copy or multicopy? Where the lines outcrossed? This is important because it is possible that the age-dependent regulation they observe could be due to factors other than the promoter sequences. For example, the unc-54 3' UTR is well known to drive gene expression in the posterior intestine, and introns in GFP stabilize the mRNA.

These details, including the primer sequences used to clone the microRNA promoters, are described fully in Martinez et al., 2008 (cited in the manuscript), which introduced the miRNA promote*r*::GFP library that we employed. In brief, the GFP and 3′ UTR were cloned from pPD95.75, the well-known “Fire Lab Vector Kit” construct containing a GFP with artificial introns and the *unc-54* 3′ UTR. The transgenes are low-copy random integrants generated via microparticle bombardment, using UNC-119 rescue as a marker. For reader convenience, we now include these details in the Materials and methods.

We outcrossed only minimally as part of breeding the reporter lines into the *spe-9(hc88)* background. We did not conduct more extensive outcrossing, as all the analysis in this work consists of within-strain comparisons which are by nature internally controlled. It is formally possible that some background allele in (say) the *unc-119(-)* strain used in the library construction in Martinez et al., confers lifespan predictive power on all GFPs, and our limited back-crossing removed this allele from half of the GFPs tested but not the other half. Beyond that unlikely circumstance, we do not see much of a route for significant confounds of the key results due to genetic background.

Several strains were additionally outcrossed as part of breeding them into other mutant/transgenic backgrounds; in no case did this further outcrossing alter the lifespan predictive properties of the transgenes. (Admittedly, had the lifespan-predictive properties of some strain in fact disappeared in the *daf-16(-)* background, say, we would then have had to carefully show that this was due to the introduction of the *daf-16(-)* allele and not the loss or gain of some other background allele. This did not occur, however.)

Nevertheless, we definitely share the reviewer’s general concerns about transgene-specific effects, and it is something that we thought a lot about as we completed this study. Overall, we believe that it is unlikely that the lifespan-predictive properties observed are caused by the introns or UTR, as those are shared across all of the transgenes we tested – while only half of the transgenes were lifespan-predictive. Moreover, despite the *unc-54* 3′ UTR, we do not observe posterior-intestine expression in most of the lifespan-predictive transgenes (Figure 1 and Table 1). In our work and that of Martinez *et al.*, these transgenes generally show independent and distinct spatial and temporal expression patterns. (We have, however, observed this problematic property of the *unc-54* 3′ UTR in other transgenic strains. In our admittedly limited experience, this issue seems to arise more in high-copy extrachromosomal transgenes and integrants derived therefrom, compared to the low-copy transgenes produced by microparticle bombardment.)

It is possible that the lifespan-predictive properties we observe require interactions between the introns/UTR and specific regulatory sequences from the transgene or integration site. Again, however, this does not conflict with or call into question our central claims.

3) The authors make several claims that need better support. They claim that age-dependent predictors are "independent of insulin signaling". But they only look at the dependency for one effector of this pathway, DAF-16, when there is extensive literature going back decades showing that SKN-1/NRF, HSF-1/HSF, and many other transcription factors also act downstream of the DAF-2 receptor (and independently of DAF-16) in the regulation of lifespan. Could those effectors of insulin signaling be involved?

The reviewer is correct: *daf-16* independence is not the same as IIS-independence, and other effectors of insulin signaling could certainly be involved. We aim to do better than to perpetuate such un-rigorous characterizations, and we thank the reviewer for this reminder.

We have amended the text to state that the predictors are independent of the *daf-16* branch of IIS rather than IIS as a whole.

4) The authors claim in the title that "Global, cell non-autonomous gene regulation drives individual lifespan." Yet this set of claims is one of many models that can explain their findings. Issues with these specific claims:4.1) Gene regulation: What is the evidence that transcriptional changes cause differences in lifespan? The authors say in the results that all their models share a "mutual transcriptional relationship with a single lifespan-determining process." Does that process need to be transcriptional just because it also affects expression of the GFP reporters?

We have revised the manuscript to call more attention to the discussion of this point, which is both critical and subtle. The observed fluorescence levels are the result of (at least) the rates of transcription, mRNA degradation, translation, GFP folding, photobleaching, unfolding, and degradation. However, after transcription, the mRNA and GFP produced by all of the transgenes in the library are identical. Thus, the processes that endows some but not all transgenes with lifespan-predictive properties must relate to transcription.

It is possible that transcriptional regulation that limits GFP expression to a certain cell type and/or temporal window could produce a lifespan-predictive reporter simply due to the fact that the rate of some downstream process (mRNA degradation, translation, etc.) is itself predictive of lifespan in that cell type / window of time. However, as noted in the manuscript, for all sites of expression and temporal windows in which we found lifespan-predictive transgenes, we also found non-predictive transgenes. As such, we conclude that transcriptional regulation *per se* is the most likely driver of lifespan predictivity.

4.2) Cell non-autonomous: I am not sure what the rationale for this claim is. Why couldn't there be a set of cell autonomous killing processes in the different tissues where the separate predictors of lifespan are expressed? This would be similar to links in a chain, the chain breaks when the weakest link breaks, but links break independently (autonomously). What would the authors have expected to turn out differently if the lifespan determining process acted cell autonomously?

We have clarified the explanations in the text of (a) why we believe the processes to be cell/tissue non-autonomous and (b) what experimental results we would have expected in the autonomous vs. non-autonomous cases.

In short, the key observation is that the expression levels of the three transgenes tested were mutually correlated / anti-correlated. The simplest explanation for correlated expression levels across different cells/tissues is a non-autonomous process.

We previously failed to state directly that the transgene levels are correlated. In the original text, we simply noted that the transgenes are statistically redundant with respect to future lifespan. While this is mathematically equivalent to a statement that their expression levels are correlated, we should have directly stated that fact and its relevance to the question of autonomy. We discuss this more carefully in the revised manuscript, and have added Figure 6—figure supplement 1 to show the correlations directly.

In more detail, transgene levels in one cell or tissue cannot be significantly correlated with levels of a different transgene in another cell or tissue without some information flow between the cells/tissues. The most likely candidate for this would be cell/tissue non-autonomous signaling. It is also formally possible that the future expression level of both transgenes could somehow be determined in or before the last common ancestor cell, and that information is subsequently inherited across cell divisions (e.g., by levels of some long-lived protein or stable regulatory process). In this way, levels of two transgenes could be correlated across different cells or tissues without any contemporaneous signaling. We cannot formally rule this out. However, the fact that the transgenes are expressed throughout life but are not lifespan-predictive until later adulthood militates somewhat against such a mechanism.

Another possibility is that whatever factors act across cells/tissues to couple transgene expression are distinct from those that act cell-autonomously to influence each transgene in proportion to future lifespan. For this to be true, there would have to be three transcriptional programs at work for any pair of reporter transgenes: one that acts non-autonomously to cause the transgenes to be correlated with one another, and two that act autonomously in each cell/tissue to independently cause the expression of each transgene to be separately correlated with future lifespan.

Of these three possible interpretations, the possibility we present in the manuscript is by far the simplest and most straightforward.

Next we address the scenario the reviewer suggests, where lifespan is set by (e.g.) three independent cell-autonomous failure processes, each reported on separately by a different transgene. In that case, there would be no correlation among the levels of the transgenes, since each failure process acts independently (and thus independently influences the levels of its reporter transgene). Similarly, each transgene would independently predict future lifespan; i.e. they would not be statistically redundant. In other words, the R^2^ value for a joint regression with multiple predictors would simply be the sum of the R^2^ values for regressing each alone. (Perhaps surprisingly, this is true regardless of whether the first failure kills the individual, or whether that requires two or all three failures.) This is more or less the opposite of what we actually observed, and that is why we infer that the reported-on processes are not independent / cell-autonomous.

5) The authors claim in the Abstract that they "demonstrate a hierarchy among several transgenes expressed in distinct tissues" that report on a single lifespan determining process. Could the transgenes with non-additive lifespan predicting capacity just have overlapping independent (ie non-hierarchical) transcriptional determinants? (that is could they share regulatory elements that each report on an independent lifespan determinant?)

We agree that this scenario, in which the reporters integrate across more or fewer independent lifespan-determining processes, is quite possible. In fact, it is explicitly depicted in Figure 6E (the model labeled “most integrative”).

We used the term “hierarchy” in the sense of an informational hierarchy: the *mir-47* reporter provides strictly more information about lifespan than the *mir-793* reporter, which itself provides strictly more information than the reporter for *mir-240-786*. While this is consistent with a true genetic hierarchy (in terms of e.g. a signaling cascade), other biological scenarios, such as the “most integrative” model above, are also consistent with the data. To illustrate this more clearly, we added another panel to Figure 6E showing the least-commitment model of the informational hierarchy: a Venn diagram with nested circles.

We believe that the use of the term “hierarchical” in the abstract is reasonable and warranted by the data, especially given the way that this claim is developed in the main text. Nevertheless, we have clarified the text; if the reviewers prefer, we can find a different term entirely.